# CLUSTER-DRIVEN ADVERSARIAL PERTURBATIONS FOR ROBUST CONTRASTIVE LEARNING

## ABSTRACT

Adversarial contrastive learning aims to learn a representation space robust to adversarial inputs using only unlabeled data. Existing methods typically generate adversarial perturbations by maximizing the contrastive loss during adversarial training. However, we find that the effectiveness of this approach is influenced by the composition of positive and negative examples in a minibatch, which is not explicitly controllable. To address this limitation, we propose a novel approach to adversarial contrastive learning, where adversarial perturbations are generated based on the clustering structure of the representation space learned through contrastive learning. Our method is motivated by the observation that contrastive learning produces a well-separated representation space, where similar data points cluster together in space, while dissimilar ones are positioned farther apart. We hypothesize that perturbations directed toward neighboring (the second nearest to be specific) clusters are likely to cross the decision boundary of a downstream classifier built upon contrastive learning, effectively acting as adversarial examples. A key challenge in our approach is to determine a sufficiently large number of clusters, for which the number of classes in the downstream task would serve the purpose but is typically unknown during adversarial contrastive learning. Therefore, we employ the silhouette score to identify the optimal number of clusters, ensuring high-quality clustering in the representation space. Compared to the existing approaches, our method achieved up to $2.25\%$ and $5.05\%$ improvements in robust accuracy against PGD and Auto-Attack, respectively, showing slight improvement in standard accuracy as well in most cases.

## 1 INTRODUCTION

The challenge of learning meaningful representations from unlabeled data remains crucial due to the scarcity of labeled data and the high cost of labeling. Self-supervised learning uses pretext tasks (e.g., jigsaw puzzles (Noroozi & Favaro, 2016) or rotation prediction (Gidaris et al., 2018)) to capture the underlying structure of unlabeled data. Among these approaches, contrastive learning (Hadsell et al., 2006; Chen et al., 2020; He et al., 2020; Chen & He, 2021) has made significant progress by pulling positive pairs closer while pushing the negative pairs farther apart in the representation space. Furthermore, the representations learned by contrastive learning achieve comparable performance with supervised learning on downstream tasks.

The success of contrastive learning has prompted interest in learning adversarially robust representations from unlabeled data. Recent works have focused on combining adversarial training with contrastive learning to enhance robustness (Zhang et al., 2022; Kim et al., 2020; Jiang et al., 2020; Fan et al., 2021; Luo et al., 2023). While adversarial training in supervised learning aims to correctly classify adversarial examples, adversarial contrastive learning achieves robustness by ensuring that adversarial and clean examples have similar representations.

In adversarial contrastive learning, generating adversarial examples is challenging due to the absence of labeled data. Existing approaches (Kim et al., 2020; Jiang et al., 2020; Fan et al., 2021; Luo et al., 2023) have addressed this challenge by maximizing the contrastive loss to generate perturbations for adversarial training. However, without explicitly considering the relationship between the resulting perturbations and the decision boundary of a downstream classifier, there is no guarantee that the created perturbations will be adversarial with respect to the decision boundary.

To address the limitation of prior approaches, we focus on the well-clustered representations learned by contrastive learning, where similar data points are closely clustered, while dissimilar points are farther apart. We conjecture that the perturbation directed toward the neighboring cluster may cross the decision boundary of a downstream classifier when the neighboring cluster is composed of dissimilar data to the original sample, given that the representations tend to be linearly separable (Chen et al., 2020; He et al., 2020; HaoChen et al., 2021; Tosh et al., 2021) and the data distribution of the downstream task is sufficiently similar to that of data used for contrastive learning. Based on this idea, we propose a novel cluster-driven adversarial contrastive learning method.

Our main contributions are:

- We propose a new adversarial contrastive learning framework utilizing the clustering structure in the representation space. Our method chooses the number of clusters by the silhouette score, eliminating the need for an extra hyperparameter tuning.
- We show by experiment that our proposed method can produce a fully robust representation, that is, adversarial linear fine-tuning makes little difference to standard linear fine-tuning.
- Our experimental results show that our proposed method outperforms the state-of-the-art methods in robust accuracy against PGD and Auto-Attack adversaries, with a slight increase in standard accuracy in the majority of cases.

## 2 RELATED WORKS

**Adversarial training.** Since the discovery of neural networks' vulnerablility to adversarial attacks (Szegedy et al., 2014), various defense methods (Madry et al., 2018; Zhang et al., 2019; Papernot et al., 2016; Xu et al., 2018; Metzen et al., 2017; Xie et al., 2019) have been proposed to enhance their adversarial robustness.

Among these defenses, adversarial training has emerged as one of the most widely adopted approaches, where adversarial examples are incorporated into the training process to improve adversarial robustness. Standard adversarial training (Madry et al., 2018) is formulated as follows:

$$\min_{f \in \mathcal{F}} \mathbb{E}_{(\tilde{x},y) \sim \mathcal{D}} \Big[ \max_{||\delta||_\infty \leq \epsilon} \mathcal{L}_{\text{CE}}(f(\tilde{x} + \delta), y) \Big].$$

Here, $\mathcal{L}_{\text{CE}}(\cdot, \cdot)$ is the cross-entropy loss, $f(\cdot)$ is an instance from a family of neural networks $\mathcal{F}$. Specifically, $f : \mathcal{X} \to \mathbb{R}^d$ maps input data $\tilde{x} \in \mathcal{X}$ to a $d$-dimensional output. The set $\mathcal{F}$ includes all possible neural networks. $(\tilde{x}, y)$ is a pair of an input and its label sampled from a data distribution $\mathcal{D}$, and $\delta$ is an adversarial perturbation.

However, there is a trade-off between standard generalization and adversarial roubstness (Tsipras et al., 2019), which leads adversarial training to reduce standard accuracy. TRADES (Zhang et al., 2019), a variant of standard adversarial training, was proposed to strike a balance between robustness and standard accuracy more effectively. TRADES optimizes the following objective:

$$\min_{f \in \mathcal{F}} \mathbb{E} \Big[ \mathcal{L}_{\text{CE}}(f(\tilde{x}), y) + \lambda \max_{||\delta||_\infty \leq \epsilon} \text{KL}(f(\tilde{x}), f(\tilde{x} + \delta)) \Big],$$

where KL denotes the KL-divergence and $\lambda > 0$ is a trade-off parameter that balances the standard accuracy and the robustness. The first term ensures that the model maintains strong performance on clean examples, while the second term encourages consistent predictions between clean and adversarial examples, thereby improving robustness.

**Contrastive learning.** Contrastive learning (Hadsell et al., 2006) is a self-supervised learning approach that learns representations from unlabeled data by encouraging the positive pairs to be similar and the negative pairs to be dissimilar in their representations. Contrastive learning optimizes neural networks $f$ and $g$ through the following objective:

$$\min_{f,g} \mathbb{E}_x \Big[ \mathcal{L}_{\text{CL}}(x, \mathcal{P}(x), \mathcal{N}(x); g, f) \Big],$$

where $x$ is the anchor sample, $\mathcal{P}(x)$ denotes a set of positive samples, and $\mathcal{N}(x)$ represents a set of negative samples. The neural network $f$ is an encoder that extracts the representation vector $z$ from

the anchor sample $x$ and $g$ is a projection head. A typical contrastive loss $\mathcal{L}_{\text{CL}}$ can be defined as follows (Chen et al., 2020):

$$\mathcal{L}_{\text{CL}}(x, \mathcal{P}(x), \mathcal{N}(x); g, f) := -\log \frac{\sum_{x_p \in \mathcal{P}(x)} \exp(\text{sim}(g \circ f(x), g \circ f(x_p))/\tau)}{\sum_{x_a \in \mathcal{N}(x) \cup \mathcal{P}(x)} \exp(\text{sim}(g \circ f(x), g \circ f(x_a))/\tau)}, \quad (1)$$

where $\tau > 0$ is a temperature parameter. For an original sample $\tilde{x}$, the positive pair $(x, x_{\text{p}}) = (t(\tilde{x}), t'(\tilde{x}))$ is built by applying transformations $t$ and $t'$ to $\tilde{x}$, respectively, where $t$ and $t'$ are randomly sampled from a pre-defined transformation set $\mathcal{T}$. Transformed versions of other samples within the mini-batch are treated as negative samples. The function $\text{sim}(u, v)$ denotes the similarity between vectors $u$ and $v$, with cosine similarity being a commonly used measure. Cosine similarity is defined as the dot product between $\ell_2$ normalized vectors $u$ and $v$. Additionally, $g \circ f$ denotes the composition of neural networks $g$ and $f$.

**Adversarial contrastive learning.** Motivated by the remarkable achievements of contrastive learning, several methods have been proposed to integrate adversarial training with contrastive learning to enhance the robustness.

Jiang et al. (2020) enforces the representations to be invariant to perturbations by performing contrastive learning on the perturbed positive pair $(x+\delta_x, x_p+\delta_p)$ instead of $(x, x_{\text{p}})$, where $\delta_x$ and $\delta_p$ are adversarial perturbations applied to the anchor sample $x$ and positive sample $x_p$, respectively. Similarly, Kim et al. (2020) proposed a robust contrastive learning (RoCL) framework that maximizes the cosine similarity between the representations of the clean (anchor) sample $x$ and the sample with a perturbation $x + \delta_x$ by including $x + \delta_x$ in the positive set $\mathcal{P}(x)$ of the contrastive loss.

Adversarial contrastive learning (ADVCL) (Fan et al., 2021) advances the RoCL by introducing the high-frequency component of the original sample $\tilde{x}$ as an additional positive sample, considering that the adversarial perturbations are primarily concentrated on the high-frequency region (Wang et al., 2020). In addition, ADVCL incorporates supervision loss using pseudo-labels generated by an additional encoder pre-trained on the ImageNet dataset (Deng et al., 2009).

Zhang et al. (2022) proposed decoupled adversarial contrastive learning (DeACL) that divides the previous adversarial contrastive learning into two stages: in stage 1, (standard) contrastive learning is adopted to train an encoder; in stage 2, adversarial training is performed with perturbations generated using the encoder pre-trained in stage 1.

Dynamic adversarial contrastive learning (DYNACL) (Luo et al., 2023) introduces a dynamic augmentation strategy on the method proposed by Jiang et al. (2020), where the strength of data augmentation is gradually reduced as the training progresses. This approach balances the need for strong augmentation in representation learning while mitigating its negative impact on robustness. DYNACL++ (Luo et al., 2023) post-processes the encoder trained with DYNACL via linear probing and adversarial full fine-tuning (Kumar et al., 2022), where pseudo-labels are generated using the encoder pre-trained with DYNACL. Xu et al. (2023a) proposed a robustness-aware coreset selection method to accelerate existing adversarial contrastive learning methods. In addition, Xu et al. (2023b) enhanced the robustness of existing adversarial contrastive learning methods (Jiang et al., 2020; Luo et al., 2023) by incorporating adversarial invariant regularization (AIR) into the adversarial contrastive loss.

## 3 METHODOLOGY

### 3.1 MOTIVATION

In this section, we discuss the limitations of existing adversarial contrastive learning methods and explain the motivation for proposing a novel perturbation generation approach.

For a given unlabeled dataset, adversarial contrastive learning aims to build an adversarially robust encoder whose learned representations preserve resistance to adversarial attacks in the downstream tasks. The existing adversarial contrastive learning methods (Kim et al., 2020; Jiang et al., 2020; Fan et al., 2021; Luo et al., 2023) generate perturbations for adversarial training by maximizing the contrastive loss as follows:

$$\delta_x^*, \{\delta_p^*\}, \{\delta_n^*\} \in \underset{\delta_x, \{\delta_p\}, \{\delta_n\}}{\arg\max} \mathcal{L}_{\text{CL}}(x + \delta_x, \mathcal{P}_\delta(x), \mathcal{N}_\delta(x); g, f),$$

Table 1: Comparison of adversarial impact between random noise and perturbations directed toward the neighboring cluster. The evaluation is performed on the amount of decrease in predicted probability for the true class (Prob. Dec.) and attack success rate (Attk. Succ.). All values are in %. Perturbations generated by our approach are denoted as $\delta_{\text{nbr}}$.

| | CIFAR-10 | | CIFAR-100 | | STL-10 | |
| | Prob. Dec. | Attk. Succ. | Prob. Dec. | Attk. Succ. | Prob. Dec. | Attk. Succ. |
| --- | --- | --- | --- | --- | --- | --- |
| random noise | 1.84 | 2.59 | 4.41 | 9.56 | 1.01 | 1.49 |
| $\delta_{\text{nbr}}$ | 15.97 | 22.08 | 36.62 | 56.74 | 25.99 | 30.66 |

where $\mathcal{P}_\delta(x) = \{x_p + \delta_p \mid x_p \in \mathcal{P}(x), \ ||\delta_p||_\infty \leq \epsilon\}$ denotes the perturbed positive set and $\mathcal{N}_\delta(x) = \{x_n + \delta_n \mid x_n \in \mathcal{N}(x), ||\delta_n||_\infty \leq \epsilon\}$ represents the perturbed negative set. As a result, the perturbed anchor sample $x + \delta_x^*$ is likely to move closer to the perturbed negative set $\mathcal{N}_\delta(x)$ (given a sufficiently large number of negative samples) and farther from the perturbed positive set $\mathcal{P}_\delta(x)$. However, the perturbation budget $\epsilon$ is often small, and therefore the perturbed positive and negative samples will remain close to their original counterparts. This implies that the generation of $\delta_x^*$ will be dominantly influenced by the positive set $\mathcal{P}(x)$ and negative set $\mathcal{N}(x)$. Consequently, the perturbation $\delta_x^*$ is likely to be adversarial if the negative samples (usually implemented using data points in a mini-batch excluding the original sample $\tilde{x}$ of an anchor point $x$) belong to different classes compared to $x$. However, there is no guarantee for such composition because mini-batch samples are selected uniformly at random, and thus, the effectiveness of created perturbations can be limited.

On the other hand, Zhang et al. (2022) proposed to generate adversarial perturbations not by maximizing the contrastive loss, but by minimizing the cosine similarity with the representations of the encoder pre-trained with SimCLR (Chen et al., 2020):

$$\delta_x^* \in \underset{||\delta_x||_\infty \leq \epsilon}{\arg\max} \, \text{CosSim}(f_\theta(x + \delta_x), f_{\text{pre}}(x)), \tag{2}$$

where $\text{CosSim}(u, v) = -\frac{u^T v}{||u|| ||v||}$ denotes the negative dot product between $\ell_2$ normalized vectors $u$ and $v$ (i.e. negative cosine similarity), $f_\theta$ is the encoder adversarially trained and $f_{\text{pre}}$ refers to the encoder pre-trained using SimCLR. However, it is unclear whether the perturbation generated by the above optimization (Eq. (2)) would be adversarial, effectively pushing $x$ to the other classes.

To address the limitations of the current approaches mentioned above, we suggest a new idea utilizing the geometrical structure of the representation space learned by contrastive learning. This representation space is defined by well-structured clusters of semantically similar data and clear distinctions between dissimilar data. Building on these properties, we propose a novel approach to generate perturbations directed toward the neighboring clusters (we choose the second nearest cluster as the neighbor based on our empirical results).

As a proof of concept for our idea, we conducted experiments to assess the adversarial effectiveness of perturbations generated toward the neighboring clusters in the representation space learned by SimCLR. For evaluation, we attached a linear classifier to the frozen encoder pre-trained with SimCLR and fine-tuned it. We then measured the probability of misclassification (attack success rate) and the amount of decrease in predicted probability for the true class induced by the perturbations. Details on the experimental procedure are available in the Appendix A.1. Table 1 compares the adversarial effectiveness of random noise and the perturbations generated by our approach. Our perturbations reduce the predicted probability for the true class by up to 25.7 times more and achieve an attack success rate by up to 20.6 times more compared to random noise. This suggests that the perturbations directed toward neighboring clusters are likely to act as adversarial examples.

## 3.2 NCP-ACL

In this section, we describe our proposal method, which we call NCP-ACL (Neighboring-Cluster Pursuit Adversarial Contrastive Learning). The overall structure of our NCP-ACL consists of two stages to make use of the clustering structure of the representation space learned through contrastive learning: (1) pre-training the encoder with standard contrastive learning; (2) performing the adversarial training based on the pre-trained encoder. We denote by $f_{\theta_1}$ the encoder pre-trained at the

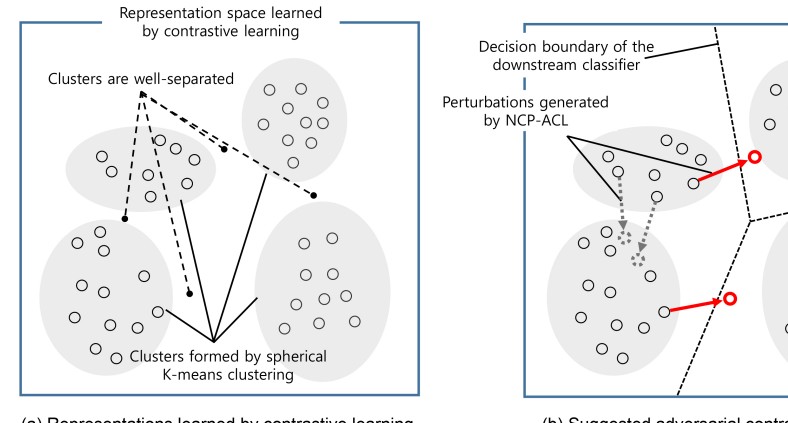

(a) Representations learned by contrastive learning    (b) Suggested adversarial contrastive learning

Figure 1: **Overview of our proposed method, NCP-ACL**. (a) In Stage 1, we obtain the representation space where clusters consisting of semantically similar data are clearly distinct. (b) In stage 2, we generate perturbations directed toward neighboring (the second nearest) clusters for adversarial training. Red arrows represent successful adversarial perturbations generated by NCP-ACL crossing the decision boundary of the downstream classifier, while grey dashed arrows indicate non-adversarial perturbations that could be generated when the neighboring cluster belongs to the same class.

first stage, and by $f_{\theta_2}$ the encoder trained at the second stage; $f_{\theta_2}$ will be our final outcome. The mechanism of our proposed method is illustrated in Fig. 1.

**Stage 1: standard contrastive learning.** We first train the encoder $f_{\theta_1}$ with SimCLR (Chen et al., 2020), one of the standard (non-robust) contrastive learning methods, which learns to locate semantically similar data points close to each other in the resulting representation space. In particular, the encoder $f_{\theta_1}$ is trained with the contrastive loss in Eq. (1).

**Stage 2: NCP adversarial training.** In the second stage, called the NCP (Neighboring-Cluster Pursuit) adversarial training, we train the adversarially robust encoder $f_{\theta_2}$ initialized with pre-trained weights from the encoder $f_{\theta_1}$. We employ the adversarial training loss from Zhang et al. (2022), a variant of TRADES (Zhang et al., 2019), which replaces both cross entropy loss and KL divergence loss with cosine similarity loss. The adversarial training loss is defined as follows:

$$\mathcal{L}_{\text{NCP-ACL}}(x; \theta_2) := \text{CosSim}(f_{\theta_2}(x), z_1) + \lambda \, \text{CosSim}(f_{\theta_2}(x^{\text{adv}}), f_{\theta_2}(x)), \tag{3}$$

where $z_1 = f_{\theta_1}(x)$ denotes the representation of the input $x$ produced by the encoder $f_{\theta_1}$ (whose weight $\theta_1$ is frozen at stage 2) and $x^{\text{adv}}$ is the adversarial input created from $x$.

The first term in Eq. (3) aims to preserve the characteristics of representation space learned at stage 1 by maximizing the cosine similarity between representation vectors of $f_{\theta_1}$ and $f_{\theta_2}$, which is related to the standard accuracy of our final model $f_{\theta_2}$. The second term is to make the representation vector of the adversarial input $x^{\text{adv}}$ similar to the representation vector of the clean example, thereby enhancing the adversarial robustness of $f_{\theta_2}$. Here, $\lambda > 0$ is a hyperparameter for balancing the standard and robust errors.

We propose to generate the adversarial input $x^{\text{adv}}$ through the following process: (i) we extract the representation vectors of training data using the encoder $f_{\theta_2}$; (ii) we apply the spherical K-means clustering (Hornik et al., 2012) to the extracted representation vectors. Since the representation space is learned using contrastive loss based on the cosine similarity, spherical K-means clustering is used instead of regular K-means; (iii) For each data point $x$, we find the centroid of the cluster it belongs to, denoted as $C_x$, and the centroid of the neighboring cluster, denoted as $C_{\text{nbr}(x)}$ (we selected the second nearest cluster as the neighbor according to our empirical results described in Appendix B.2); (iv) finally, we create the adversarial perturbation $\delta^*$ as a solution to the following optimization:

$$\delta^* \in \underset{||\delta||_\infty \leq \epsilon}{\arg\max} \, \{\text{CosSim}(f_{\theta_2}(x + \delta), C_x) - \text{CosSim}(f_{\theta_2}(x + \delta), C_{\text{nbr}(x)})\}, \tag{4}$$

---

**Algorithm 1** Stage 2: NCP adversarial training algorithm

---

**Input**: A dataset $\mathbb{D}$, a pre-trained encoder $f_{\theta_1}$, the number of training epochs $N$, the number of training steps $T$, and a hyperparameter $\lambda > 0$.
**Output**: an adversarially robust encoder $f_{\theta_2}$.
**Initialize** $f_{\theta_2}$ with the pre-trained weights of $f_{\theta_1}$.

   **for** epoch $\in \{0, \ldots, N-1\}$ **do**
       Perform the spherical K-means clustering on $\mathbb{D}$.
       Compute cluster centroids $\{\tilde{C}_k\}_{k=1}^{K}$.
       **for** step $\in \{0, \ldots, T-1\}$ **do**
          Sample a mini-batch $B = \{x_0, \ldots, x_{m-1}\}$
          **for** $i = 0, \ldots, m-1$ (in parallel) **do**
             $z_i \leftarrow f_{\theta_1}(x_i)$.
             $C_{x_i} \leftarrow$ the nearest centroid to $x_i$ in $\{\tilde{C}_k\}_{k=1}^{K}$.
             $C_{\text{nbr}(x_i)} \leftarrow$ the second nearest (neighboring) centroid to $x_i$.
             $x_i^{\text{adv}} \leftarrow x_i + \delta_i^*$, where $\delta_i^*$ is a solution to Equation (4).
          **end for**
          Update $\theta_2$ by minimizing $\sum_{i=0}^{m-1} \mathcal{L}_{\text{NCP-ACL}}(x_i)$.
       **end for**
   **end for**

---

for which we used the 5-step projected gradient descent (PGD) with a perturbation budget of $\epsilon = 8/255$. Here, the first term directs the perturbation away from the cluster to which the data point $x$ belongs, aiming to enhance the adversarial effect by pushing the data point away from the distribution of similar data. The effect of the first term is validated through an ablation study in the Appendix B.1. The second term guides the perturbation toward the neighboring cluster.

The resulting adversarial input $x^{\text{adv}} = x + \delta^*$ is expected to be located farther from $C_x$ and closer to $C_{\text{nbr}(x)}$ compared to the original input $x$ in the representation space. As illustrated in the right side of Fig. 1, depending on the relative position of the neighboring cluster and the decision boundary of the downstream classifier in the representation space, the generated perturbations can be either adversarial (red arrows) or non-adversarial (gray dashed arrows). In particular, perturbations are likely to be non-adversarial, when the data point is located between the neighboring cluster and the decision boundary. In this case, we expect $x^{\text{adv}}$ to act as a positive sample in regular contrastive learning, since the perturbation is directed toward a cluster composed of data from the same class as the data point $x$ in the downstream task. The procedure of the stage 2 is summarized in Algorithm 1.

## 4 EXPERIMENTS

In this section, we evaluate the effectiveness of our proposed method, NCP-ACL, against the current state-of-the-art (SOTA) methods, including ADVCL (Fan et al., 2021), DeACL (Zhang et al., 2022), DYNACL (Luo et al., 2023), DYNACL++ (Luo et al., 2023) and DYNACL-AIR++ (Xu et al., 2023b). The comparison focuses on the quality of the encoder representations learned by each method, providing an in-depth analysis of adversarial robustness achieved by adversarial contrastive methods. Unless otherwise noted, we reported the average results over five runs with different random seeds.

### 4.1 EXPERIMENTAL SETUP

**Datasets and model architecture.** We evaluate the performance of our proposed method and the SOTA approaches on three standard benchmark datasets for adversarial contrastive learning: CIFAR-10 (Krizhevsky & Hinton, 2009), CIFAR-100 (Krizhevsky & Hinton, 2009), and STL-10 (Coates et al., 2011). Consitent with prior works (Zhang et al., 2022; Kim et al., 2020; Jiang et al., 2020; Fan et al., 2021; Luo et al., 2023; Xu et al., 2023a;b), we adopt the ResNet-18 (He et al., 2016) as the encoder architecture for the evaluation.

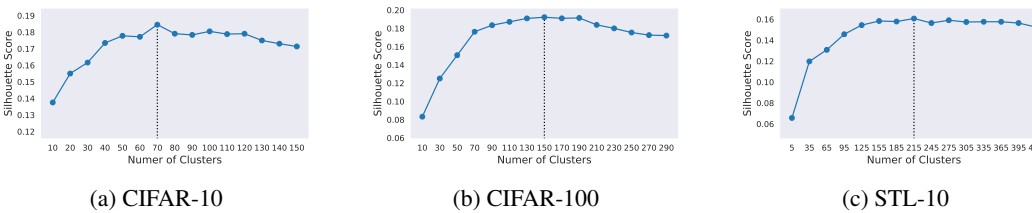

(a) CIFAR-10        (b) CIFAR-100        (c) STL-10

Figure 2: Silhouette scores with different numbers of clusters $K$ on CIFAR-10, CIFAR-100, and STL-10. The optimal $K$ values, corresponding to the highest silhouette scores, are 70, 150, and 215 for CIFAR-10, CIFAR-100, and STL-10, respectively. Due to the space constraint, we only present the results with the random seed set to 0.

Table 2: Performance comparison of adversarial contrastive learning methods on CIFAR10, CI-FAR100, and STL10 datasets. Standard linear fine-tuning (SLF) has been applied for performance evaluation in terms of standard accuracy (SA), robust accuracy (RA), and Auto-Attack accuracy (AA). The best and the second best cases are marked with boldface and underlines, resp. 'n/a' indicates that the method is not applicable due to the absence of a required information.

| Method | CIFAR-10 | | | CIFAR-100 | | | STL-10 | | |
| | SA(%) | RA(%) | AA(%) | SA(%) | RA(%) | AA(%) | SA(%) | RA(%) | AA(%) |
|---|---|---|---|---|---|---|---|---|---|
| ADVCL | 80.85 | 50.45 | 42.57 | 48.34 | 27.67 | 19.78 | n/a | n/a | n/a |
| DeACL | 80.17 | 53.95 | 45.31 | 52.79 | 30.74 | 20.34 | 80.87 | 65.86 | 58.50 |
| DYNACL | 75.38 | 48.48 | 45.56 | 45.85 | 22.76 | 19.43 | 69.61 | 48.76 | 47.17 |
| DYNACL++ | 79.79 | 49.37 | 47.05 | **53.93** | 21.97 | 19.71 | 70.97 | 49.91 | 47.80 |
| DYNACL-AIR++ | 81.86 | 49.01 | 47.12 | 53.90 | 23.20 | 20.69 | 71.36 | 50.23 | 48.42 |
| NCP-ACL (ours) | **82.02** | **54.06** | **49.22** | 51.00 | **32.01** | **23.32** | **84.06** | **68.11** | **63.55** |

**Fine-tuning strategies.** To compare the performance of adversarial contrastive learning methods, we attach a linear classification layer on top of the respective final encoder learned by each method. To train the linear classification layer on the frozen encoder, we consider two types of fine-tuning strategies: standard linear fine-tuning (SLF) and adversarial linear fine-tuning (ALF). In SLF, the linear classification layer is trained with original training examples and their corresponding ground-truth labels, while in ALF, it is trained with adversarial examples generated by the 10-step $\ell_\infty$-PGD attack (Madry et al., 2018) with a perturbation budget of $\epsilon = 8/255$ and a step size of $\alpha = 2/255$.

**Evaluation metrics.** Performance evaluation is conducted using three metrics: standard accuracy (SA), robust accuracy (RA), and Auto-Attack accuracy (AA). Each corresponds to accuracy on clean data, adversarial examples generated by 20-step $\ell_\infty$-PGD attack with $\epsilon = 8/255$ and $\alpha = 2/255$, and adversarial examples from Auto-Attack (Croce & Hein, 2020) with $\epsilon = 8/255$ in $\ell_\infty$-norm, respectively.

**Selection of the number of clusters.** In stage 2 of our proposed NCP-ACL, the number of clusters $K$ is required as a hyperparameter for spherical K-means clustering. With unknown class numbers in the downstream task, we determine the value of $K$ using the silhouette score (Rousseeuw, 1987), a metric for evaluating clustering quality. We compute silhouette scores by clustering representations extracted from the initial encoder $f_{\theta_2}$, and fix the optimal $K$ value throughout training. This approach eliminates the need for additional hyperparameter tuning to select the number of clusters, thus significantly reducing the computational cost. Figure 2 shows silhouette scores for varying $K$ on CIFAR-10, CIFAR-100, and STL-10 with the random seed fixed at 0. Here, the optimal $K$ values are 70, 150, and 215 for CIFAR-10, CIFAR-100, and STL-10, respectively. Additional details are provided in Appendix A.2.

### 4.2 PERFORMANCE EVALUATION OF ADVERSARIAL CONTRASTIVE LEARNING

**Evaluation of representation quality via standard linear fine-tuning.** To compare the quality of representations learned by adversarial contrastive learning methods, we first report the performance results using the standard linear fine-tuning (SLF) as the fine-tuning strategy. Table 2 shows the

Table 3: The effect of different fine-tuning strategies: SLF and ALF. In both, adversarial contrastive learning is performed to learn a robust encoder. In SLF, an attached linear classifier on top of the encoder is fine-tuned with clean labeled samples. In ALF, the linear classifier is fine-tuned instead with PGD-based adversarial training. The results are obtained using CIFAR-10.

| Method | SLF | | | ALF | | |
|---|---|---|---|---|---|---|
| | SA (%) | RA (%) | AA(%) | SA (%) | RA (%) | AA(%) |
| ADVCL | 80.85 | 50.45 | 42.57 | 80.00 | 51.93 | 44.65 |
| DeACL | 80.17 | 53.95 | 45.31 | 78.81 | **54.61** | 46.09 |
| DYNACL | 75.38 | 48.48 | 45.56 | 72.89 | 49.66 | 46.01 |
| DYNACL++ | 79.79 | 49.37 | 47.05 | 78.82 | 51.18 | 48.46 |
| DYNACL-AIR++ | 81.86 | 49.01 | 47.12 | 79.59 | 51.77 | 48.66 |
| NCP-ACL (ours) | **82.02** | **54.06** | **49.22** | **80.40** | **54.61** | **49.23** |

performance comparison across CIFAR-10, CIFAR-100, and STL-10, with the best and the second-best results highlighted in boldface and underlines, respectively (results of ADVCL and DeACL for CIFAR-10 and CIFAR-100 were taken from the corresponding papers). Across all three datasets, NCP-ACL consistently achieved the best performance in terms of standard accuracy (SA), robust accuracy (RA), and Auto-Attack accuracy (AA) except for SA on CIFAR-100. In the case of CIFAR-100, ours showed about a 3% drop in SA compared to the best performer of the case, DYNACL++. We believe that it was a trade-off made by our method to achieve about 10% and 4% boost in RA and AA, resp., compared to the performance numbers of DYNACL++, and one can probably find an operating point of ours to improve SA further, perhaps sacrificing RA and AA slightly. For example, as shown in Fig. 3, setting $K = 10$ instead of our $K = 150$ (based on the best silhouette score) increases SA by about 1% to 51.76% while RA and AA decrease to 30.66% and 21.77%, respectively. However, despite this decrease, both RA and AA still significantly outperform DynACL++. (The results for AdvCL on STL-10 are unavailable: AdvCL requires pseudo-labels generated by an additional encoder pre-trained on ImageNet, which were not provided. Despite generating pseudo-labels ourselves, we were unable to achieve comparable performance.) The experimental result indicates that our proposed method is likely to produce more robust representations, demonstrating the effectiveness of our approach as an adversarial contrastive learning strategy.

**Evaluation of representation quality via adversarial linear fine-tuning.**  In Table 3, we compare the performance of adversarial contrastive methods on CIFAR10 with the adversarial linear fine-tuning (ALF). Note that ALF serves as additional robust training for the linear classifier attached on top of the encoders trained through adversarial contrastive learning, where adversarial inputs generated by the PGD attack are used. We can see that our proposal method NCP-ACL outperforms all competitors in the ALF setting as well. Also, the performance numbers of NCP-ACL with ALF have not changed much from those with SLF, indicating that the extra robust training by ALF may not be necessary for our method.

### 4.3 IMPACT OF THE NUMBER OF CLUSTERS ON PERFORMANCE

Our method uses spherical K-means clustering for generating perturbations for adversarial training. Therefore, the method's sensitivity to the choice of $K$ values could be an issue. Therefore, we analyzed the relationship between the number of clusters and the performance of NCP-ACL regarding standard accuracy (SA), robust accuracy (RA), and Auto-Attack accuracy (AA). In Fig. 3, the plots show how these performance metrics change due to the choice of the number of clusters $K$. The red striped bars show the performance with the best $K$ value for the silhouette score, which we chose for our experiments.

Overall, the results show that when the value of $K$ increases over a threshold (e.g., $K \geq 20$ for CIFAR-10), the RA and AA values are not very sensitive to the choice of $K$. In addition, we can achieve near-best performance by using the $K$ value with the highest silhouette score, where the experiments are performed with this $K$ value.

We guess that when $K$ is large enough, the clusters' boundaries of data from different classes tend to align with the decision boundaries of downstream classifiers. When $K$ is larger than necessary,

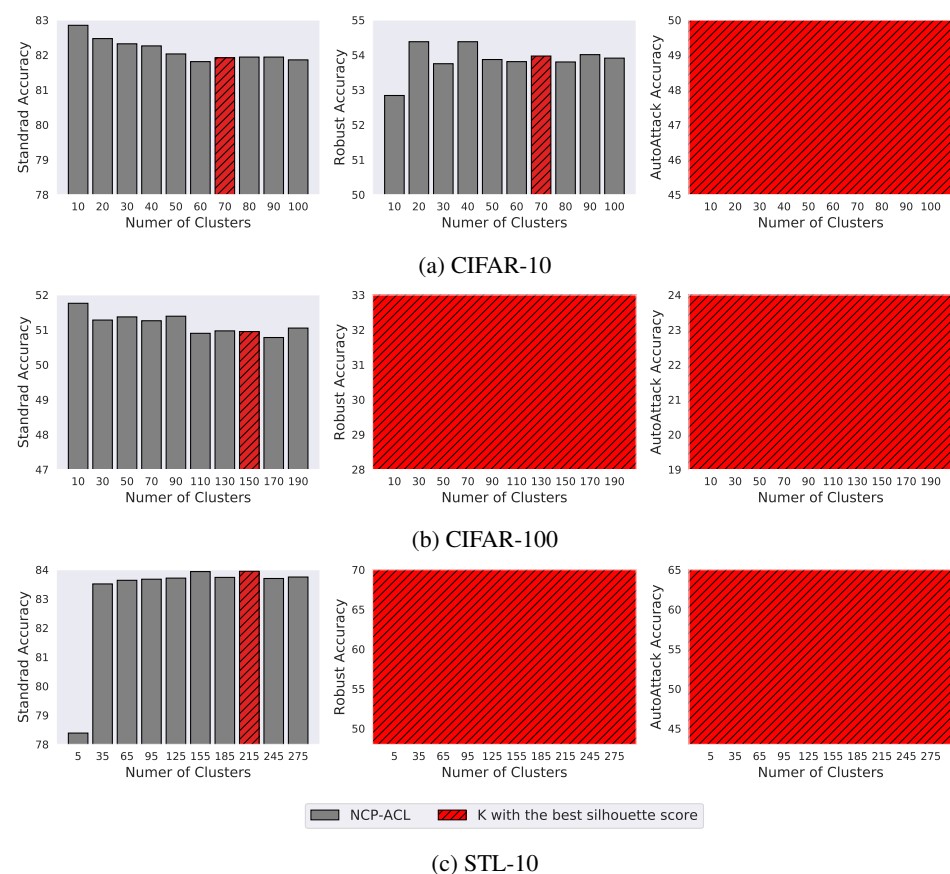

(a) CIFAR-10

(b) CIFAR-100

(c) STL-10

Figure 3: The relation between prediction performance and the number of clusters $K$ in NCP-ACL. The accuracy with the $K$ value that achieved the highest silhouette score is marked in the red striped pattern (the results with the random seed set to 0 are shown).

we conjecture that the clustering will likely form more clusters within each class. This may result in the perturbations created by NCP-ACL as non-adversarial since the second nearest clusters may become more likely to consist of the data from the same class as the nearest clusters. The tendency of standard accuracy drops with the increment of $K$ in CIFAR-10 and CIFAR-100 might reflect the phenomenon where non-adversarial augmentations work against classifying regular inputs. Still, the effect of $K$ being large does not seem significant, at least in the range of $K$ values we tried.

## 5 CONCLUSION

In this paper, we proposed a novel adversarial contrastive learning method called NCP-ACL. We tried to address the problem in the current methodologies where adversarial examples are typically constructed by maximizing the contrastive learning loss, and as a result, the generated perturbation may not be adversarial with respect to the decision boundary obtained through fine-tuning. In NCP-ACL, we utilized the well-clustered representations learned by contrastive learning, where we generated adversarial inputs by directing the perturbations toward the neighboring clusters. In this way, we expected that the perturbations would be toward the direction of crossing the decision boundary of a downstream classifier. Through the experiments, we showed that our proposal outperforms the current state-of-the-art methods in learning robust representations, leading to higher standard and robust predictions after fine-tuning.

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

## A EXPERIMENTAL DETAILS

### A.1 SETUP FOR PROOF OF CONCEPT EXPERIMENT

To validate whether perturbations generated toward neighboring clusters act adversarially, we conduct a proof-of-concept experiment. The overall process consists of five steps: (i) We train the encoder using SimCLR. (ii) We perform spherical K-means clustering on the representations learned in step (i), where the resulting clusters are expected to consist of similar data points. Here, the number of clusters $K$ is selected using the silhouette score. (iii) For each data point, we generate a perturbation directed toward the centroid of the neighboring cluster. (iv) For evaluating the adversarial effectiveness of generated perturbations, we attach a linear classifier to the frozen encoder pre-trained in step (i) and train the linear classifier using labeled training samples. (v) Using the model from step (iv), we measure the attack success rate and average decrease in prediction probability for the true class caused by perturbations.

We employ the ResNet18 architecture for the encoder. Perturbations are generated by optimizing Eq. (4) without the first term $\text{CosSim}(f(x + \delta), C_x)$, where we use 5-step PGD with a perturbation budget of $\epsilon = 8/255$. To precisely measure the adversarial effect of perturbations, we evaluate only on test samples correctly classified by the model from step (4). From experimental results presented in Table 1, we confirm that the perturbations generated by our approach are likely to cross the decision boundary of the downstream classifier.

### A.2 PRE-TRAINING SETTINGS

#### A.2.1 STAGE 1: STANDARD CONTRASTIVE LEARNING

We use the solo-learn (da Costa et al., 2022), a library of self-supervised methods, for the SimCLR implementation. Following Chen et al. (2020), we train the model for 1000 epochs with a batch size of 512. For CIFAR-10 and CIFAR-100, we adopt the LARS optimizer with momentum 0.9, weight decay 1e-4, and cosine learning rate decay with a 10-epoch linear warm-up. For STL-10, weight decay is set to 1e-5. The temperature parameter in contrastive loss is set to 0.5. For data augmentation, we use a random resized crop with a size of 32 and a scale between 0.08 and 1.0, color jitter with a strength of 0.5 and a probability of 0.8, grayscale with a probability of 0.2, Gaussian blur with a probability of 0.5, and horizontal flip with a probability of 0.5 for CIFAR-10 and CIFAR-100. For STL-10, we only adjust the size parameter of the random resized crop to 96 and the strength of the color jitter to 1.0.

#### A.2.2 STAGE 2: NCP ADVERSARIAL TRAINING

We employ the spherical K-mean clustering implementations provided by kmcuda (kmc, 2019) and AFK-MC$^2$ (Bachem et al., 2016) to improve speed and scalability. kmcuda is a GPU-accelerated K-means implementation, while AFK-MC$^2$ is a simple and fast seeding algorithm for K-means clustering. The number of clusters for spherical K-means clustering was determined using the silhouette score. The optimal $K$ values across different random seeds are 70, 60, 70, 90, and 75 for CIFAR-10, 150, 125, 130, 85, and 110 for CIFAR-100, and 210, 145, 155, 200, and 175 for STL-10. In the case of STL-10, the training dataset consists of $5,000$ labeled samples and $100,000$ unlabeled samples, where unlabeled samples include other types of animals (bears, rabbits, etc.) and vehicles (trains, buses, etc.) not found in the labeled set. This explains why the best $K$ value for STL-10 is considerably larger than its number of classes, 10.

In stage 2, the model is trained for 200, 100, and 1000 epochs on CIFAR-10, CIFAR-100, and STL-10, respectively. For other hyperparameters, we follow the settings of DeACL (Zhang et al., 2022), using a batch size of 256 and a learning rate of 0.5. For data augmentation, as in DeACL, we apply weak augmentation comprised of random crop (after padding) to the original image size (32 for CIFAR-10 and CIFAR-100, 96 for STL-10) and horizontal flip with a probability of 0.5. We set $\lambda$ to 6. In addition, we adopt the dual batch normalization method from Jiang et al. (2020), which utilizes separate batch normalization layers: one for clean examples and the other for adversarial examples.

Table 4: The effect of different perturbation generation strategies in adversarial contrastive learning. Strategy (i) generates perturbations away from the data point's own cluster, strategy (ii) generates perturbations toward the neighboring cluster, and strategy (iii) is our approach, combining both strategies.

|  | CIFAR-10 | | | CIFAR-100 | | | STL-10 | | |
|---|---|---|---|---|---|---|---|---|---|
|  | SA(%) | RA(%) | AA(%) | SA(%) | RA(%) | AA(%) | SA(%) | RA(%) | AA(%) |
| strategy (i) | 82.29 | 52.06 | 47.77 | 50.82 | 31.08 | 22.18 | 83.76 | 65.70 | 61.12 |
| strategy (ii) | **83.46** | 52.05 | 47.16 | **52.93** | 31.05 | 22.26 | 83.86 | 61.06 | 55.40 |
| strategy (iii) (ours) | 82.02 | **54.06** | **49.22** | 51.00 | **32.01** | **23.32** | **84.06** | **68.11** | **63.55** |

## A.3 FINE-TUNING SETTINGS

### A.3.1 STANDARD LINEAR FINE-TUNING

We train the linear classification layer with 25 epochs and a batch size of 512 using clean examples, where the encoder pre-trained using NCP-ACL is fixed. We adopt SGD optimizer with momentum 0.9 and weight decay 2e-4. The learning rate is set to 0.1 on CIFAR-10 and 10.0 for CIFAR-100 and STL-10. The learning rate decays by 10 times at epochs 15 and 20.

### A.3.2 ADVERSARIAL LINEAR FINE-TUNING

For adversarial linear fine-tuning, we train the linear classification layer for 25 epochs using adversarial examples generated by the 10-step $\ell_\infty$-PGD attack with $\epsilon = 8/255$ and $\alpha = 2/255$. We set the learning rate to 0.1 on CIFAR-10. In addition, we use the SGD optimizer and learning rate decay with the same settings as the standard linear fine-tuning.

## B ADDITIONAL EXPERIMENTS

### B.1 ABLATION STUDY ON PERTURBATION GENERATION STRATEGIES

This section compares the effectiveness of different perturbation generation strategies in adversarial contrastive learning. We focus on three distinct perturbation generation strategies derived from components of Eq. (4): (i) generating perturbations directed away from the cluster to which each data point belongs, (ii) generating perturbations directed toward the neighboring cluster, and (iii) generating perturbations that are directed toward the neighboring cluster while simultaneously moving away from the data point's own cluster. Strategy (i) corresponds to maximizing the first term in Eq. (4), strategy (ii) corresponds to minimizing the second term in Eq. (4), and strategy (iii) uses the entire equation for creating perturbations as employed in our method.

Using the two-stage framework of NCP-ACL, we perform adversarial contrastive learning with perturbations generated by the three strategies described above. To assess the robustness of the learned representations, we apply standard linear fine-tuning to the adversarially trained encoder. Table 4 presents the performance comparison across CIFAR-10, CIFAR-100, and STL-10. The results show that combining strategies (i) and (ii) generates the most effective perturbations for adversarial training, thereby achieving the best robust accuracy.

### B.2 EFFECT OF TARGETING DIFFERENT NEIGHBORING CLUSTERS FOR ADVERSARIAL CONTRASTIVE LEARNING

In our NCP-ACL, we generate perturbations that are directed away from the cluster to which the data point belongs and simultaneously toward the neighboring cluster. Specifically, we used the second nearest cluster as the neighboring cluster for perturbation generation. In this section, we conduct experiments to analyze the impact of targeting different neighboring clusters on the quality of leaned representations. That is, we compared the performance of adversarial contrastive learning when directing perturbations toward the farther neighboring clusters instead of the second nearest one.

Table 5: The impact of targeting different neighboring clusters for adversarial contrastive learning. Targeting the second closest neighboring cluster ($n = 2$) was used in our method, and performance is compared when perturbations are targeted to the third ($n = 3$), fourth ($n = 4$), and fifth ($n = 5$) closest neighboring clusters.

|  | CIFAR-10 | | | CIFAR-100 | | | STL-10 | | |
|---|---|---|---|---|---|---|---|---|---|
|  | SA(%) | RA(%) | AA(%) | SA(%) | RA(%) | AA(%) | SA(%) | RA(%) | AA(%) |
| $n = 2$ (NCP-ACL) | 82.02 | 54.06 | 49.22 | 51.00 | 32.01 | 23.32 | 84.06 | 68.11 | 63.55 |
| $n = 3$ | 81.73 | 54.41 | 49.93 | 51.06 | 32.32 | 23.64 | 83.50 | 68.38 | 64.25 |
| $n = 4$ | 81.87 | 53.91 | 49.85 | 51.16 | 31.91 | 23.63 | 83.47 | 68.01 | 63.67 |
| $n = 5$ | 82.09 | 54.11 | 49.72 | 51.14 | 31.61 | 23.41 | 83.17 | 67.79 | 63.39 |

Using the same two-stage framework as in Appendix B.1, we performed adversarial training with perturbations generated toward different neighboring clusters. From Table 5, we observe little performance difference across these variations – therefore, we simply set the second nearest cluster ($n = 2$) as the neighboring cluster.

