# OpenReview forum: "Cluster-Driven Adversarial Perturbations for Robust Contrastive Learning"
_ICLR.cc/2025/Conference — Submitted to ICLR 2025_

### Official Review · Reviewer_TbcU · 2024-10-28

**Soundness:** 3
**Presentation:** 4
**Contribution:** 3
**Rating:** 6
**Confidence:** 5

**Summary:**

The paper proposes a novel approach to adversarial contrastive learning, leveraging clustering within the representation space. The method, named Neighboring-Cluster Pursuit Adversarial Contrastive Learning (NCP-ACL), generates adversarial perturbations by directing them toward neighboring clusters in the learned representation space. This technique aims to address limitations in existing adversarial contrastive learning methods by making perturbations more effective in crossing decision boundaries in downstream classifiers. The paper demonstrates significant improvements in robust accuracy against adversarial attacks compared to state-of-the-art methods.

**Strengths:**

- The paper addresses a notable limitation in current adversarial contrastive learning techniques, where perturbations may not be effective in crossing decision boundaries. By utilizing the clustering structure in the representation space, the proposed NCP-ACL method generates more meaningful perturbations.

- The experimental results on CIFAR-10, CIFAR-100, and STL-10 datasets show that NCP-ACL outperforms existing methods in terms of both robust and standard accuracy. The approach yields consistent improvements across different datasets and settings, with up to 5.05% improvement in robust accuracy against Auto-Attack.

- The use of silhouette scores to automatically determine the optimal number of clusters is a practical approach that simplifies hyperparameter tuning.

**Weaknesses:**

- The paper should discuss the computational cost introduced by the additional clustering step. It would be helpful to provide an analysis of how the clustering affects training time compared to baseline methods.
- The method’s reliance on spherical K-means clustering could pose scalability challenges for high-dimensional representation spaces.
- The paper would benefit from a discussion on the theoretical implications or guarantees provided by the method. For instance, can the clustering structure always ensure that perturbations cross decision boundaries effectively, and under what conditions?

**Questions:**

1. How does the clustering step, including the calculation of silhouette scores, affect the overall computational time? Is there a significant increase in training time compared to other adversarial contrastive learning methods?

2. What assumptions are made regarding the quality of the clustering in the representation space? Could the proposed method perform poorly if the representation space is not well-clustered, and if so, under what specific conditions might this occur?

3. The paper shows results for standard and adversarial linear fine-tuning. Have you tried adversarial full fine-tuning (AFF)?

---

> ### Author Response · Authors · 2024-11-25
>
> We sincerely thank Reviewer TbcU for their thoughtful feedback and valuable suggestions.
>
> > **Response to W1&Q1**
>
> To evaluate the computational cost of the additional clustering step in NCP-ACL, we compared its total training time with other SOTA methods. As shown in Table 1, NCP-ACL achieves competitive training time. It achieves the second shortest training time for CIFAR-10 and CIFAR-100. For STL-10, its training time is slightly longer than the second shortest (DYNACL), but the difference is minimal, demonstrating comparable efficiency.
>
> Table 2 provides a detailed breakdown of NCP-ACL's training time. The clustering step, including silhouette score calculation and K-means clustering, accounts for less than 16\% of the total training time across all datasets, taking $0.97$ hours for CIFAR-10, $0.68$ hours for CIFAR-100, and $9.42$ hours for STL-10. This indicates that the clustering step introduces only minor overhead.
>
> Importantly, the clustering step significantly enhances the adversarial robustness. This trade-off is well-justified by the superior robust accuracy achieved by NCP-ACL. In future work, we would reduce this cost by optimizing the clustering process with parallel computation or more efficient algorithms.
>
> **Table 1. Comparison of total training time between SOTA methods and NCP-ACL.**
> (The best and the second-best results highlighted in **bold** and *italic*, respectively)
>
> | **Method**       | **CIFAR-10 (hours)** | **CIFAR-100 (hours)** | **STL-10 (hours)** |
> |-------------------|--------------------|---------------------|------------------|
> | **ADVCL**        | 48.12              | 47.67               | 74.34           |
> | **DeACL**        | **8.00**           | **7.99**            | **43.99**       |
> | **DYNACL**       | 32.93              | 34.26               | _56.07_         |
> | **DYNACL++**     | 33.72              | 35.40               | 57.36           |
> | **DYNACL-AIR++** | 43.98              | 44.69               | 71.66           |
> | **NCP-ACL (ours)** | _10.53_           | _8.75_              | 59.99           |
>
> **Table 2. Breakdown of NCP-ACL training time by component.**
>
> | **Component**               | **CIFAR-10 (hours)** | **CIFAR-100 (hours)** | **STL-10 (hours)** |
> |-----------------------------|--------------------|---------------------|------------------|
> | **Model Training**          | 9.56              | 8.07               | 50.57           |
> | **Silhouette Score Calculation** | 0.43              | 0.40               | 2.82            |
> | **K-means Clustering**      | 0.54              | 0.28               | 6.60            |
> | **Total Training Time**     | 10.53             | 8.75               | 59.99           |
>
>
> > **Response to Q2**
>
> According to [1], encoders trained with contrastive learning generally produce a representation space where similar data points are positioned closely, naturally forming well-clustered structures. However, the quality of clustering in such representation space could be degraded depending on several factors:
> - If the encoder lacks sufficient capacity, it may fail to effectively capture the underlying structure of the data, resulting in poorly separated clusters within the representation space.
> - If the data used for contrastive learning contains noise,  imbalanced classes, or insufficient diversity, the quality of clustering would be poor. These issues can lead to overlapping or ambiguous clusters, which hinder the formation of distinct and well-structured representations.
>
> In these cases, the perturbations generated by NCP-ACL may fail to effectively induce adversarial impact, reducing the robustness of adversarial training.
>
> [1] T. Wang and P. Isola, ''Understanding Contrastive Representation Learning through Alignment and Uniformity on the Hypersphere,'' ICML'20
>
> > **Response to Q3**
>
> We conducted experiments to evaluate AFF, where both the encoder and the linear classifier are fine-tuned using adversarial training. The results are shown in the table below. Notably, NCP-ACL outperforms other SOTA methods under AFF in both robust accuracy (RA) and adversarial accuracy (AA) while maintaining competitive standard accuracy (SA).
>
> **Performance comparison of adversarial contrastive methods under adversarial full fine-tuning (AFF) on CIFAR-10.**
>
> | **Method**      | **SA (%)** | **RA (%)** | **AA (%)** |
> |------------------|------------|------------|------------|
> | **ADVCL**       | 83.62      | 52.77      | 49.77      |
> | **DeACL**       | 83.95      | 54.18      | 50.39      |
> | **DYNACL**      | 81.71      | 54.20      | 50.76      |
> | **DYNACL++**    | 81.95      | 54.12      | 51.01      |
> | **DYNACL-AIR++**| 82.44      | 53.84      | 50.46      |
> | **NCP-ACL (ours)** | **84.01**  | **55.05**  | **51.75**  |

---

> > ### Comment · Reviewer_TbcU · 2024-12-01
> >
> > Thank you for your detailed rebuttal and for considering adversarial full fine-tuning (AFF) as suggested. I have decided to maintain my score.

---

> > > ### Author Response · Authors · 2024-12-02
> > >
> > > Thank you for your response. We would like to provide additional answers to address points that were not covered in our previous reply.
> > >
> > > For the second weakness, as noted in our response to W1 and Q1, we have mitigated scalability issues arising from K-means clustering by utilizing a GPU-accelerated implementation. This approach significantly reduces computational overhead, ensuring the scalability of our method.
> > >
> > > Regarding the third weakness, our method relies on three key assumptions to generate effective adversarial examples:
> > >   - Representation space structure: In the representation space learned through contrastive learning, similar data points are positioned closely. This property, supported by prior work [1], results in a well-clustered structure when appropriate clustering is applied.
> > >   - Task similarity: The spatial structure of similarity and dissimilarity among input data in contrastive learning is sufficiently similar to that of the downstream task, ensuring well-aligned cluster structures.
> > >   - Well-separated dataset: In the input space of the downstream task, data points from the same class are close to each other, while data points from different classes are located far apart.
> > >
> > > Under these assumptions, NCP-ACL performs clustering with $K$ that maximizes the silhouette score, thereby increasing the likelihood that each cluster consists of data points from the same class.
> > >
> > > In the NCP-ACL method, perturbations are generated by moving away from the nearest cluster while targeting a neighboring cluster. Clusters formed with the highest silhouette score are likely to group data points from the same class tightly. Thus, perturbations that move away from the regions where data points of the same class are densely gathered increase the likelihood of crossing decision boundaries and creating adversarial examples. Simultaneously, directing perturbations toward the neighboring clusters on the data manifold helps prevent deviations from the underlying data manifold, ensuring that adversarial examples remain realistic and meaningful.
> > >
> > > We acknowledge that these assumptions may not hold perfectly, as contrastive learning is performed without labeled data. However, this approach allows us to effectively generate adversarial examples using unlabeled data, supported by our experimental results.
> > >
> > > Please feel free to leave a comment if you have any additional questions or concerns.
> > >
> > > [1] T. Wang and P. Isola, ''Understanding Contrastive Representation Learning through Alignment and Uniformity on the Hypersphere,'' ICML'20

---

### Official Review · Reviewer_bfjU · 2024-11-03

**Soundness:** 2
**Presentation:** 2
**Contribution:** 2
**Rating:** 6
**Confidence:** 4

**Summary:**

The paper introduces a new adversarial contrastive learning framework that leverages the clustering structure in the representation space, using the silhouette score to select the optimal number of clusters without extra hyperparameter tuning. Additionally, the authors demonstrate that their method achieves fully robust representations, where adversarial fine-tuning has minimal impact compared to standard fine-tuning, and outperforms state-of-the-art methods in robust accuracy against PGD and Auto-Attack, with a slight improvement in standard accuracy. However, some concerns about writing and methods remain as follows.

**Strengths:**

1.	The authors propose to use cluster-driven adversarial perturbations to improve model’s robustness, such framework makes fully use of the spherical K-means method on contrastive learning.
2.	They empirically show that perturbations generated through the method are harder for models to process compare to random perturbations.
3.	They conduct extensive experiments on CIFAR10, CIFAR100, and STL-10 for classification tasks and robustness evaluation, which verify the effectiveness of the proposed method.

**Weaknesses:**

1.	The motivation for this work is not clear. The authors claim that “the effectiveness of this approach is influenced by the composition of positive and negative examples in a minibatch, which is not explicitly controllable.”. However, what is the effect of positive and negative examples in a minibatch? What is the relation of the adversarial perturbations to positive and negative examples?
2.	I am concerned about the significance of this work. Traditional contrastive learning can obtain a model with high standard accuracy, but if adversarial samples are added to the training, will it lead to a significant decrease in standard accuracy like traditional adversarial training [A][B], even though robustness is improved? If this problem exists, can we achieve more friendly standard accuracy and/or robust accuracy by combining adversarial purification [C][D] and adversarial detection [E][F] after obtaining a standard model through traditional contrastive learning?
3.	The experiments in Table 1 are not convincing to me, comparing only random noise is insufficient. Noise generated under the same paradigm[G][H][I] should be compared, otherwise, the improvement cannot be attributed to breaking the decision boundary.
4.	In Experiment 4.2, the statement “We believe that it was a trade-off” lacks supporting evidence. Additional experiments or academic references are needed to substantiate this claim.
5.	The experiments in Table 3 focus on the small dataset (CIFAR-10) and do not demonstrate the upper limit of the proposed method's generalization ability. Could more complex datasets be considered?

**Questions:**

1.	In Lines 52-53 and 193, why do you want to push the adversarial sample to other classes for sure? Is there some supported theory or experiment?
2.	Why the key challenge in the approach is to determine a sufficiently large number of clusters? What is its effect? It seems that the results in Fig. 3 show that we can obtain a modest performance if we take a cluster number greater than 30.
3.	What is the motivation of employing the silhouette score?
4.	For the adversarial contrastive learning, what kind of encoder is considered to have well-structured clusters (Line 196)? What impact would it have on your method if the clustering ability itself is not very good?
5.	What the advantages of Eqn. (4) compared with Eqn. (2) in theory?
6.	In Section 3.1, the rationale for choosing the second-nearest neighbor lacks clarity— is there a theoretical explanation beyond empirical justification?
7.	In the introduction, the assumption that "there is no guarantee that the created perturbations will be adversarial with respect to the decision boundary" is intuitive. Could this assumption be validated through some experiments?
8.	The experiments in Table 4 show an improvement in robustness, but a decrease in standard accuracy on the CIFAR-10/100 datasets. Can a reasonable explanation be provided for this?
9.	As I understand it, each cluster should have a first-nearest neighbor cluster. If that's the case, why is the n=1 data missing in the ablation experiments in Table 5?

**Minor issues**
1. The concept “adversarial linear fine-tuning” and “standard linear finetuning” are referenced before definition.
2. In Lines 262 and 269, Should the encoder used for generating adversarial samples be $f_{\theta_1}$?
3. In line 20, page 2, "vulnerablility" should be "vulnerability".
4. In line 31, page 2, "roubstness" should be "robustness".
5. In line 27, page 6, "Consitent" should be "Consistent".
6. The section on contrastive learning could be strengthened by incorporating additional related papers, such as [J-M].
7. Would it make better if samples of different classes in Figure 1 were labeled with different colors?

**Reference**

[A]	Towards deep learning models resistant to adversarial attacks. ICLR 2018.

[B]	Improving robustness using generated data. NeurIPS 2021.

[C]	Diffusion models for adversarial purification. ICML 2022.

[D]	Adversarial purification with score-based generative models. ICML 2021.

[E]	Detecting adversarial data by probing multiple perturbations using expected perturbation score. ICML 2023.

[F]	 A practical bayesian approach to adversarial detection. CVPR 2021.

[G]	Decoupled adversarial contrastive learning for self-supervised adversarial robustness. ECCV 2022.

[H]	 Rethinking the effect of data augmentation in adversarial contrastive learning. ICLR 2023.

[I]	Enhancing adversarial contrastive learning via adversarial invariant regularization. NeurIPS 2023.

[J]	Intra- and Inter-Slice Contrastive Learning for Point Supervised OCT Fluid Segmentation. IEEE Trans. Image Process. 2022.

[K]	Unsupervised Domain Adaptation Via Contrastive Adversarial Domain Mixup: A Case Study on COVID-19. IEEE Trans. Emerg. Top. Comput. 2024.

[L]	Cycle contrastive adversarial learning with structural consistency for unsupervised high-quality image deraining transformer. Neural Networks, 2024.

[M]	 Semi-Supervised Dual-Stream Self-Attentive Adversarial Graph Contrastive Learning for Cross-Subject EEG-based Emotion Recognition. IEEE Trans. Affective Comput. 2024.

---

> ### Author Response · Authors · 2024-11-25
>
> We sincerely thank you for your detailed review and insightful feedback on our paper.
>
> > **Response to W1**
>
> In contrastive learning, the composition of the minibatch determines the organization of the positive and negative sets. As described in lines 113-116, the positive set consists of augmented views of the same data point used in creating the anchor sample, while the negative set comprises all other data points in the minibatch.
>
> In existing adversarial contrastive learning methods, perturbations are generated by maximizing the contrastive loss, which moves the perturbed anchor closer to the negative set and farther from the positive set. Consequently, the effectiveness of this approach heavily depends on the composition of the negative set.
>
> Since minibatch composition is determined by random sampling, situations can arise where the negative samples primarily belong to the same class as the anchor sample. In such cases, the perturbations are directed toward the same class regions, failing to exhibit adversarial natures. Additionally, if the negative set contains samples from diverse classes, the perturbation's effect is dispersed across multiple regions, lacking a specific adversarial direction. As a result, perturbations generated by existing methods are likely to be less effective.
>
> > **Response to W2**
>
> While adversarial detection and purification methods offer valuable alternatives for defending against adversarial attacks, they are not foolproof. Adversarial examples that bypass these mechanisms can still compromise the models. Therefore, it is essential to develop robust models that can inherently resist adversarial attacks. Additionally, techniques specifically designed to circumvent detection and purification highlight the limitations of relying solely on such defenses (e.g., [1] and [2]).
>
> This need for inherent robustness has driven active research in adversarial contrastive learning, as demonstrated by works such as ADVCL, DeACL, DynACL, and DynACL-AIR. These methods aim to directly enhance the model's robustness against adversarial attacks without depending on external defense mechanisms.
>
> Moreover, adversarial detection and purification methods often require training additional models and introduce preprocessing overhead, which can significantly increase computational and time costs. By contrast, adversarial contrastive learning integrates robustness directly into the model during training, providing a more efficient and practical approach to adversarial defense.
>
> [1] N. Carlini and D. Wagner, ''Adversarial examples are not easily detected: Bypassing ten detection methods,'' AISec 2017.
>
> [2] M. Kang, D. Song, and B. Li, ''DiffAttack: Evation attacks against diffusion-based adversarial purification,'' NeurIPS 2023
>
> > **Response to W3&Q7**
>
> The proof-of-concept experiment (Table 1) comparing random noise with perturbations directed toward the second-nearest cluster was intended to validate whether perturbations targeting the second-nearest cluster have an adversarial impact. Random noise served merely as a baseline to confirm that the perturbations we generated exhibit adversarial natures. It was not designed to demonstrate the superiority of our method. In the first general response, we present the comparative experiments evaluating the effectiveness of
> perturbations crafted by NCP-ACL and SOTA methods. Please refer to it.
>
> > **Response to W4**
>
> The trade-off between adversarial robustness and standard accuracy is a well-recognized phenomenon in the literature. For instance, [1] and [2] demonstrate that improving adversarial robustness comes at the cost of reduced standard accuracy. In Experiment 4.2, the observed trade-off is consistent with these findings, suggesting that improving adversarial robustness may inherently affect standard accuracy.
>
> [1] Tsipras, D., Santurkar, S., Engstrom, L., Turner, A., and Madry, A., ''Robustness may be at odds with accuracy,'' ICLR 2019.
>
> [2] Zhang, H., Yu, Y., Jiao, J., Xing, E., Ghaoui, L. E., and Jordan, M,, ''Theoretically principled trade-off between robustness and accuracy,'' ICML 2019.

---

> > ### Author Response · Authors · 2024-11-25
> >
> > > **Response to W5**
> >
> > We conducted additional experiments using more complex datasets, specifically CIFAR-100 and STL-10, under the ALF setting. The results are presented below. When considering both SLF and ALF collectively, NCP-ACL consistently demonstrates superior performance across all datasets, achieving the highest RA and AA. These findings further validate the generalization ability of our method beyond smaller datasets like CIFAR-10, highlighting its effectiveness across diverse and more complex datasets.
> >
> > **Performance comparison of adversarial contrastive learning methods under different fine-tuning strategies across multiple datasets.**
> > (The best and the second-best results highlighted in **bold** and *italic*, respectively)
> >
> > | Method            | Finetuning Method                   | CIFAR-10          |                  |                  | CIFAR-100        |                  |                  | STL-10           |                  |                  |
> > |--------------------|-------------------------------------|-------------------|------------------|------------------|------------------|------------------|------------------|------------------|------------------|------------------|
> > |                    |                                     | SA (%)    &nbsp;&nbsp;&nbsp;&nbsp;&nbsp;&nbsp;&nbsp;&nbsp;&nbsp;       | RA (%)    &nbsp;&nbsp;&nbsp;        | AA (%)     &nbsp;&nbsp;&nbsp;      | SA (%)      &nbsp;&nbsp;&nbsp;&nbsp;&nbsp;&nbsp;&nbsp;&nbsp;&nbsp; &nbsp;&nbsp;&nbsp;&nbsp;     | RA (%)    &nbsp;&nbsp;&nbsp;       | AA (%)    &nbsp; &nbsp; &nbsp;        | SA (%)     &nbsp; &nbsp; &nbsp; &nbsp; &nbsp; &nbsp; &nbsp;       | RA (%)      &nbsp; &nbsp; &nbsp;      | AA (%)    &nbsp; &nbsp; &nbsp;        |
> > | **ADVCL**          | Standard Linear Finetuning (SLF)   | 80.85             | 50.45            | 42.57            | 48.34            | 27.67            | 19.78            | n/a              | n/a              | n/a              |
> > | **DeACL**          |                                     | 80.17             | 53.95            | 45.31            | 52.79            | *30.74*          | 20.34            | 80.87            | 65.86            | 58.50            |
> > | **DYNACL**         |                                     | 75.38             | 48.48            | 45.56            | 45.85            | 22.76            | 19.43            | 69.61            | 48.76            | 47.17            |
> > | **DYNACL++**       |                                     | 79.79             | 49.37            | 47.05            | **53.93**        | 21.97            | 19.71            | 70.97            | 49.91            | 47.80            |
> > | **DYNACL-AIR++**   |                                     | *81.86*           | 49.01            | 47.12            | *53.90*          | 23.20            | 20.69            | 71.36            | 50.23            | 48.42            |
> > | **NCP-ACL (ours)** |                                     | **82.02**         | *54.06*          | *49.22*          | 51.00            | **32.01**        | **23.32**        | **84.06**        | *68.11*          | *63.55*          |
> > | **ADVCL**          | Adversarial Linear Finetuning (ALF)| 80.00             | 51.93            | 44.65            | 47.08            | 27.99            | 20.19            | n/a              | n/a              | n/a              |
> > | **DeACL**          |                                     | 78.81             | **54.61**        | 46.09            | 42.74            | 28.87            | 20.77            | 80.65            | 66.92            | 57.87            |
> > | **DYNACL**         |                                     | 72.89             | 49.66            | 46.01            | 43.63            | 25.57            | 20.79            | 67.83            | 50.85            | 48.43            |
> > | **DYNACL++**       |                                     | 78.82             | 51.18            | 48.46            | 51.35            | 26.88            | 22.74            | 69.68            | 51.35            | 48.50            |
> > | **DYNACL-AIR++**   |                                     | 79.59             | 51.77            | 48.66            | 51.91            | 27.10            | *23.09*          | 70.55            | 52.08            | 49.84            |
> > | **NCP-ACL (ours)** |                                     | 80.40             | **54.61**        | **49.23**        | 44.98            | 30.54            | 22.66            | *83.88*          | **70.29**        | **64.81**        |

---

> > > ### Author Response · Authors · 2024-11-25
> > >
> > > > **Response to Q1**
> > >
> > > In this work, we follow the definition of adversarial examples from [1], where they are described as imperceptible, non-random perturbations added to original images that alter a neural network’s predictions. The primary goal of adversarial contrastive learning is to train an encoder that enhances robustness against adversarial attacks in downstream tasks. This requires generating adversarial examples that cross the decision boundary of the downstream task classifier, as these boundary-crossing examples are essential for adversarial training to improve adversarial robustness ([2]). Lines 52-54 and 193 point out a limitation in existing adversarial contrastive learning methods, which may fail to generate such adversarial examples effectively, thereby reducing their effectiveness in achieving the primary goal of enhancing adversarial robustness.
> > >
> > > [1] C. Szegedy, W. Zaremba, I. Sutskever, J. Bruna, D. Erhan, I. J. Goodfellow, and R. Fergus, ''Intriguing properties of neural networks,'' ICLR 2014.
> > >
> > > [2] I. J. Goodfellow, J. Shlens, and C. Szegedy, ''Explaining and harnessing adversarial examples,'' ICLR 2015.
> > >
> > > > **Response to Q2**
> > >
> > > The number of clusters is crucial for determining clustering quality in the representation space. If the number of clusters is too small, data points from multiple classes may group into the same cluster, reducing the effectiveness of perturbations moving away from the nearest cluster to which each data point belongs. Adversarial impact is maximized when perturbations move away from clusters composed of similar data points, whereas clusters containing data from diverse classes dilute this effect.
> > >
> > > As shown in Fig. 3, performance stabilizes once the number of clusters exceeds a sufficiently large threshold. Beyond this threshold, modest variations in the number of clusters have minimal impact, highlighting the need for selecting a sufficiently large number of clusters that exceeds this threshold. The sufficiently large number of clusters depends on factors such as the dataset and structure of the representation space. For example, in NCP-ACL experiments on CIFAR-10, CIFAR-100, and STL-10, using a cluster number greater than 30 provides stable performance. However, this threshold may vary for other datasets. To address this variability, we use the silhouette score, which adaptively identifies the optimal number of clusters, ensuring clusters that are both well-formed and maintain high purity by grouping closely related data points.
> > >
> > > > **Response to Q3**
> > >
> > > The Silhouette score measures the quality of clustering by comparing each data point's average distance to points within its own cluster (intra-cluster distance) with its average distance to points in the nearest neighboring cluster (inter-cluster distance). A higher score indicates that data points are well-matched to their own clusters while being distinctly separated from others, representing meaningful and well-defined clustering. As demonstrated in [1] and [2], the Silhouette Score serves as a reliable proxy for assessing clustering quality in the absence of ground truth labels.
> > >
> > > [1] P. J. Rousseeuw, ''Silhouettes: A graphical aid to the interpretation and validation of cluster
> > > analysis,'' Journal of Computational and Applied Mathematics 1987.
> > >
> > > [2] S. C. Lowe, J. B. Haurum S. Oore, T. B. Moeslund, and G. W. Taylor, ''Zero-shot Clustering of Embeddings with Pretrained and Self-Supervised Learnt Encoders,'' NeurIPS Workshop on Robustness of Few-shot and Zero-shot Learning in Foundation Models 2023.
> > >
> > > > **Response to Q4**
> > >
> > > According to [1], encoders trained with contrastive learning generally produce a representation space where similar data points are positioned closely, naturally forming well-clustered structures. Therefore, an encoder with sufficient capacity is expected to have strong clustering ability. However, the quality of clustering in such representation space could be degraded depending on several factors:
> > > - If the encoder lacks sufficient capacity, it may fail to effectively capture the underlying structure of the data, resulting in poorly separated clusters within the representation space.
> > > - If the data used for contrastive learning contains noise,  imbalanced classes, or insufficient diversity, the quality of clustering would be poor. These issues can lead to overlapping or ambiguous clusters, which hinder the formation of distinct and well-structured representations.
> > >
> > > In these cases, similar data points may not be grouped in the representation space. Consequently, perturbations moving away from the nearest cluster may fail to induce an effective adversarial impact. This could reduce the effectiveness of adversarial training and limit robustness.
> > >
> > > [1] T. Wang and P. Isola, ''Understanding Contrastive Representation Learning through Alignment and Uniformity on the Hypersphere,'' ICML'20

---

> > > > ### Author Response · Authors · 2024-11-25
> > > >
> > > > > **Response to Q5**
> > > >
> > > > As highlighted in the first general response, Eqn. (4) achieves a significant improvement over Eqn. (2) by addressing the problem of adversarial examples deviating from the manifold. Specifically, it introduces an additional term that directs perturbations toward neighboring clusters, ensuring that the generated adversarial examples remain closer to the data manifold. This adjustment improves the robustness of adversarial training.
> > > >
> > > > Additionally, while Eqn. (2) generates perturbations based on the fixed representations learned by the SimCLR encoder, Eqn. (4) adapts perturbations to move away from the dynamically updated representations of the adversarially trained encoder. By aligning perturbations with the continuously evolving state of the encoder, Eqn. (4) maximizes the adversarial effectiveness, producing stronger and more meaningful adversarial examples.
> > > >
> > > > > **Response to Q6**
> > > >
> > > > We have provided a detailed explanation regarding the rationale for directing perturbations toward the second-nearest cluster in the second general response. Please refer to it.
> > > >
> > > > > **Response to Q8**
> > > >
> > > > As explained in the response to W4, there is an inherent trade-off between adversarial robustness and standard accuracy. This trade-off explains why, with strategy (iii), an improvement in adversarial robustness on CIFAR-10/100 comes at the cost of a decrease in standard accuracy compared to strategies (i) and (ii).
> > > >
> > > > STL-10, however, represents an exception case, which can be attributed to the structure of the representation space learned through contrastive learning. On STL-10, the representation space appears to have been structured such that adversarial examples generated using strategy (iii) distort the features less significantly compared to those generated by strategies (i) and (ii). As a result, strategy (iii) achieves simultaneous improvements in both adversarial robustness and standard accuracy on STL-10.
> > > >
> > > > Given that the primary goal of adversarial contrastive learning is to train models that are robust against adversarial attacks, we chose to employ strategy (iii) for its superior performance in enhancing robustness across datasets.
> > > >
> > > > > **Response to Q9**
> > > >
> > > > There seems to be some confusion regarding the terminology used in our method. In our proposed approach, the "nearest cluster" refers to the cluster a data point belongs to. Our method generates perturbations that move away from this nearest cluster while directing them toward the second-nearest cluster (which may have been referred to as the "first-nearest neighbor cluster" in the question).
> > > >
> > > > In Table 5 of the appendix, we compare the performance when perturbations target the second, third, fourth, and fifth nearest clusters while simultaneously moving away from the nearest cluster. To avoid further confusion, we will clarify this terminology more explicitly in future revisions. For a detailed explanation of why perturbations target the second nearest cluster, please refer to the second general response.
> > > >
> > > > > **Minor issues**
> > > >
> > > > We sincerely thank the reviewer for pointing out these minor issues and for the constructive suggestions.
> > > >
> > > > In future revisions, we will address all typographical errors and ensure that concepts such as “adversarial linear fine-tuning” and “standard linear fine-tuning” are clearly defined before being referenced, improving the clarity of the manuscript.
> > > >
> > > > Additionally, we will strengthen the section on contrastive learning by incorporating the recommended related works to provide a more comprehensive context. As part of these revisions, we will also update Figure 1 as suggested, enhancing its visual clarity.
> > > >
> > > > Thank you again for these helpful suggestions.

---

> ### Comment · Reviewer_bfjU · 2024-11-27
> **Thanks for the response**
>
> Thanks for your detailed response to the review. I appreciate the effort you have made to address the comments, however, I still have the following questions:
>
> 1.	While the authors have explained the importance of adversarial contrastive learning, the method sacrifices significant standard accuracy in exchange for a minor increase in robustness, which seems disproportionate given the additional computational cost. In contrast, methods like adversarial purification [C][D] and adversarial detection [E][F], although potentially requiring additional models, maintain model accuracy. This raises concerns about the practical significance of this work.
>
> 2.	The paper in the current vision lacks experimental results comparing the proposed method with noise generated by other baselines under the same paradigm of adversarial contrastive learning. These comparisons are crucial to validate the motivation of this work and should be included in the main tables.
>
> 3.	The authors have not provided results on larger datasets, which limits the validation of the method's effectiveness. Datasets such as CIFAR-100 and STL-10 were mentioned in the original paper.
>
> 4.	While the authors discussed and responded to many points, their revisions and discussions are not adequately reflected in the updated paper. It is important that all clarifications and modifications are incorporated into the manuscript.
>
> In summary, the current manuscripts require significant improvement.

---

> > ### Author Response · Authors · 2024-12-02
> >
> > Thank you, Reviewer bfjU, for your valuable and insightful feedback.
> >
> > > **Response to Q1**
> >
> > We acknowledge adversarial detection and purification methods can maintain standard accuracy. However, they introduce significant computational overhead during both training and inference. This makes them unsuitable for real-time applications such as autonomous driving or fraud detection, where rapid predictions are essential and additional latency is unacceptable. Furthermore, adversarial detection methods may produce false positives, misclassifying benign inputs as adversarial. Such errors can block the processing of legitimate inputs, potentially leading to severe disruptions in areas like medical diagnosis or financial transactions.
> >
> > Our work targets scenarios where only unlabeled data is available. In such cases, the performance of adversarial detection and purification methods would degrade, as they often rely on labeled data to train auxiliary models or set detection thresholds. Consequently, these methods may fail to effectively handle adversarial examples. To address such unhandled adversarial examples, it is crucial to strengthen the inherent robustness of the model using adversarial contrastive learning.
> >
> > In summary, adversarial contrastive learning offers a scalable approach by embedding robustness directly into the model, avoiding the inference-time overhead associated with detection and purification methods. In addition, the limitations of adversarial purification and detection in unlabeled data scenarios highlight the importance of continued research in this area. While there is a trade-off in standard accuracy, this approach remains valuable for real-time and resource-constrained applications, underscoring the practical significance of our work in advancing adversarial defenses.
> >
> > > **Response to Q2&Q4**
> >
> > Comparisons between perturbations generated by our proposed method and those from the SOTA adversarial contrastive learning methods are thoroughly addressed in the first general response. These results validate the motivation behind our work and clearly demonstrate the effectiveness of our method. Furthermore, we recognize the importance of such comparative experiments in highlighting the contributions of our approach. The insightful feedback and valuable suggestions from the reviewers have significantly enhanced the clarity and depth of our work. To further improve the manuscript, we plan to incorporate the aforementioned comparisons along with discussions held during the discussion period into a revised version. While it is not currently possible to upload a revised PDF, we will ensure that a revised version reflecting these improvements is submitted if an opportunity is provided after the discussion period.
> >
> > > **Response to Q3**
> >
> > It seems we misinterpreted the reviewer's comment. We initially understood W5 (the fifth weakness) as a request for additional experiments on CIFAR-100 and STL-10 using adversarial linear fine-tuning (ALF), as Table 3 only included results on CIFAR-10 with ALF. Therefore, we presented additional experimental results on these datasets. However, after reviewing the third question in the new comments, we now realize that the reviewer intended to request evaluations of our method on larger datasets such as ImageNet.
> >
> > We agree that results on larger datasets would provide valuable insights into the robustness of our approach. However, adversarial contrastive learning methods are inherently computationally intensive, which has led most studies in this field, including ours, to focus on relatively small-scale datasets such as CIFAR-10, CIFAR-100, and STL-10. For instance, ADVCL, one of the most computationally intensive methods, requires over 70 hours to train on STL-10 using a single NVIDIA 3090 RTX GPU. Due to time constraints, we are unable to conduct experiments on larger datasets during this discussion period. Nevertheless, we plan to evaluate NCP-ACL on larger datasets in future work to thoroughly demonstrate its generalization in diverse settings.

---

> ### Comment · Reviewer_bfjU · 2024-12-02
>
> Thanks your very
>  detailed response. I have increased my score from 5 to 6.

---

### Official Review · Reviewer_E4Lt · 2024-11-03

**Soundness:** 2
**Presentation:** 3
**Contribution:** 2
**Rating:** 5
**Confidence:** 4

**Summary:**

This paper introduces an adversarial contrastive learning framework called NCP-ACL (Neighboring-Cluster Pursuit Adversarial Contrastive Learning). The framework leverages the cluster structure of the representation space learned through standard contrastive learning to generate adversarial examples that challenge the decision boundary. Building on a pretrained standard contrastive learning encoder, the method identifies each data point's cluster centroid and its second-nearest neighboring cluster to create adversarial perturbations. Experimental results across three downstream tasks demonstrate the effectiveness of the proposed approach.

**Strengths:**

- This paper introduces related work and the background knowledge in a clear way.
- Experimental setting is clear and apparent.

**Weaknesses:**

- Lack of theoretical / empirical supporting arguments for claims.
   - Why the second nearest neighbor is used instead of the nearest neighbor to generate adversaries? Even in the appendix, there is only comparison using between second, third, and fourth nearest neighbors.
   - The theoretical explanation of why choosing K with the highest Silhouette scores is good for clustering.
   - The main motivation of this work is existing adversarial contrastive learning works don't guarantee that the generated adversarial examples are cross the decision boundary. Then, with the fine-tuned linear classifier, can you compare the success rate that how much generated adversarial examples are cross the decision boundary for both baseline methods and yours?
   - Why do you leverage 5-PGD during training which is relatively weaker than existing standard PGD attack with more attack steps? Since using weaker adversaries lead to a better clean accuracy, evaluating models with much stronger attack seems to be essential.
- Need more experiments with diverse recent attacks (e.g., PGD-100, CW, LGV, SPSA, DeepFool)

**Questions:**

- Is this method robust against diverse pretrained encoder with different learning algorithms such as BYOL or BarlowTwins?

---

> ### Author Response · Authors · 2024-11-25
>
> Thank you, Reviewer E4Lt, for your valuable and insightful feedback.
>
> >**Response to W1-1**
>
> There seems to be some confusion regarding the terminology used in our method. In our proposed approach, the "nearest cluster" refers to the cluster a data point belongs to. Our method generates perturbations that move away from this nearest cluster while directing them toward the second-nearest cluster (referred to as the "nearest neighbor" in the question).
>
> In Table 5 of the appendix, we compare the performance when perturbations target the second, third, fourth, and fifth nearest clusters, ensuring they move away from the nearest cluster. To avoid further confusion, we will clarify this terminology in future revisions. For a detailed explanation of why perturbations target the second nearest cluster, please refer to the second general response.
>
> >**Response to W1-2**
>
> The Silhouette score measures the quality of clustering by comparing each data point's average distance to points within its own cluster (intra-cluster distance) with its average distance to points in the nearest neighboring cluster (inter-cluster distance). A higher score indicates that data points are well-matched to their own clusters while being distinctly separated from others, representing meaningful and well-defined clustering. As demonstrated in [1] and [2], the Silhouette Score serves as a reliable proxy for assessing clustering quality in the absence of ground truth labels.
>
> [1] P. J. Rousseeuw, ''Silhouettes: A graphical aid to the interpretation and validation of cluster
> analysis,'' Journal of Computational and Applied Mathematics 1987.
>
> [2] S. C. Lowe, J. B. Haurum S. Oore, T. B. Moeslund, and G. W. Taylor, ''Zero-shot Clustering of Embeddings with Pretrained and Self-Supervised Learnt Encoders,'' NeurIPS Workshop on Robustness of Few-shot and Zero-shot Learning in Foundation Models 2023.
>
> >**Response to W1-3**
>
> We have addressed this point in the first general response, where we compare the effectiveness of perturbations generated by our method with those from baseline methods. For a detailed explanation, please refer to it.
>
> >**Response to W1-4**
>
> We use 5-PGD during training to strike a balance between computational efficiency and adversarial robustness, a standard practice in existing SOTA adversarial contrastive learning methods. While stronger attacks, such as PGD-20 or PGD-100, generate harder adversarial examples, they significantly increase computational costs, making them less practical for large-scale adversarial training. Moreover, as shown in our response to W2, models trained with 5-PGD achieve robust performance against more challenging attacks, including PGD-100, AutoAttack, and CW, with competitive robust accuracy.
>
> >**Response to W2**
>
> We thank the reviewer for suggesting the evaluation of adversarial contrastive learning methods against diverse adversarial attacks. To assess the methods under stronger attack scenarios, we focused on white-box scenarios considered more challenging than black-box scenarios. Specifically, we evaluated the robust accuracy of adversarial contrastive learning methods against PGD-20, PGD-100, AutoAttack, CW, and DeepFool on the CIFAR-10 dataset. The results are presented in the table below.
>
> The table demonstrates that NCP-ACL outperforms SOTA methods in robustness across all attacks while also achieving the highest standard accuracy. This indicates that our proposed method enables the learning of more adversarially resistant representation space compared to other approaches.
>
> **Performance comparison of the adversarial contrastive methods under various attacks on CIFAR-10.**
> (The best and the second-best results highlighted in **bold** and *italic*, respectively)
>
> | Method           | SA (%)        | AutoAttack (%) | PGD-20 (%)    | PGD-100 (%)   | DeepFool (%)  | CW (%)         |
> |-------------------|---------------|----------------|---------------|---------------|---------------|----------------|
> | ADVCL            | 80.85         | 42.57          | 50.45         | 49.30         | 44.88         | 45.77          |
> | DeACL            | 80.17         | 45.31          | _53.95_       | _53.10_       | 47.47         | 48.20          |
> | DYNACL           | 75.38         | 45.56          | 48.48         | 48.17         | 46.44         | 46.80          |
> | DYNACL++         | 79.79         | 47.05          | 49.37         | 49.00         | _47.98_       | 48.31          |
> | DYNACL-AIR++     | _81.86_       | _47.12_        | 49.01         | 48.70         | 47.88         | _48.84_        |
> | NCP-ACL (ours)   | **82.02**     | **49.22**      | **54.06**     | **53.67**     | **50.26**     | **50.91**      |

---

> > ### Author Response · Authors · 2024-11-25
> >
> > > **Response to Q1**
> >
> > To evaluate the robustness of our method against diverse pretrained encoders, we conducted experiments using BarlowTwins, BYOL, MoCo V2, and SimCLR at Stage 1 of NCP-ACL, as presented in the table below.
> >
> > NCP-ACL achieves strong performance across all encoders in terms of SA, RA, and AA, demonstrating its generalizability. While SimCLR achieves the highest RA and AA, BYOL and MoCo V2 also perform competitively. This suggests that the robustness of NCP-ACL is not dependent on a specific encoder but rather on its ability to leverage the clustering structure inherent in various contrastive learning methods.
> >
> > **Performance of NCP-ACL with various pretrained encoders on CIFAR-10.**
> >
> > | Method         | SA (%)        | RA (%)        | AA (%)        |
> > |----------------|---------------|---------------|---------------|
> > | BarlowTwins    | 78.40         | 52.56         | 46.03         |
> > | BYOL           | 80.52         | 53.27         | 47.50         |
> > | MoCo V2        | **83.66**     | 53.51         | 49.03         |
> > | SimCLR (ours)  | 82.02         | **54.06**     | **49.22**     |

---

> > > ### Comment · Reviewer_E4Lt · 2024-11-27
> > >
> > > Thank you for your detailed response.
> > > However, the theoretical explanation for choosing K with the highest Silhouette scores is not convincing to me. Also, experimental results that show how much generated adversarial examples cross the decision boundary are not enough. It would be good to analyze the prediction distribution of the correct label and incorrect label for the adversarial examples.
> > >
> > > Therefore, I maintain my score.

---

> > > > ### Author Response · Authors · 2024-12-02
> > > >
> > > > We sincerely thank the reviewer for taking the time to provide thoughtful feedback and constructive comments.
> > > >
> > > > > **Explanation for Choosing K with the Highest Silhouette Score in NCP-ACL**
> > > >
> > > > We acknowledge that our initial explanation for choosing $K$ with the highest silhouette score was insufficient. Below, we provide additional clarification to better justify our approach.
> > > >
> > > > The silhouette score is a metric used to evaluate the clustering quality by assessing how similar each data point is to its own cluster compared to the nearest neighboring cluster. Formally, the silhouette score $s(i)$ for each data point $i$ is calculated as:
> > > >
> > > > $\begin{aligned}
> > > > s(i) = \frac{b(i)-a(i)}{\max(a(i), b(i))}
> > > > \end{aligned}$
> > > >
> > > >   - $a(i)$ is the average distance between $i$ and all other points in the same cluster, measuring how tightly grouped the point is with others in the same cluster:
> > > >
> > > >     $a(i) = \frac{1}{|C_{i}|-1}\sum_{j \in C_{i}, j \neq i}d(i,j),$
> > > >     with $C_{i}$ representing the cluster to which the data point $i$ belongs, and $d(i,j)$ denoting the distance between points $i$ and $j$.
> > > >   - $b(i)$ is the average distance from $i$ to all points in the nearest neighboring cluster, quantifying the dissimilarity between $i$ and the closest cluster other than $C_i$:
> > > >
> > > >     $b(i)=\min_{C_k \neq C_i}\frac{1}{|C_k|}\sum_{j \in C_k} d(i,j),$
> > > >     where $C_k$ is any cluster other than $C_i$.
> > > >
> > > > The overall silhouette score for the dataset, $S$, is computed as the mean of the silhouette scores across all data points:
> > > >
> > > > $S = \frac{1}{N}\sum_{i=1}^{N}s(i)$, where $N$ is the total number of data points.
> > > >
> > > > When the silhouette score is maximized, the following properties are achieved:
> > > >   - $a(i)$ tends to be minimized: Data points within the same cluster are tightly grouped, reducing intra-cluster variability.
> > > >   - $b(i)$ tends to be maximized: Clusters are well-separated, increasing the inter-cluster distances.
> > > >
> > > > This optimal configuration ensures that clusters consist of highly similar data points while distinct clusters are composed of dissimilar data points. Consequently, clusters are more likely to represent semantically meaningful groupings.
> > > >
> > > > In our method, we rely on three key assumptions:
> > > >   - Representation space structure: In the representation space learned through contrastive learning, similar data points are positioned closely. This property, supported by prior work [1], results in a well-clustered structure when appropriate clustering is applied.
> > > >   - Task similarity: The spatial structure of similarity and dissimilarity among input data in contrastive learning is sufficiently similar to that of the downstream task, ensuring well-aligned cluster structures.
> > > >   - Well-separated dataset: In the input space of the downstream task, data points from the same class are close to each other, while data points from different classes are located far apart.
> > > >
> > > > Given these assumptions, performing clustering with $K$ that maximizes the silhouette score ensures that each cluster is highly likely to consist of data points from the same class. This is because the maximization process leverages the inherent structure of the representation space, which groups similar data points closely while keeping dissimilar clusters well-separated.
> > > >
> > > > In the NCP-ACL method, perturbations are generated by moving away from the nearest cluster while targeting a neighboring cluster. Clusters formed with the highest silhouette score are likely to group data points from the same class tightly. Thus, perturbations that move away from the regions where data points of the same class are densely gathered increase the likelihood of crossing decision boundaries and creating adversarial examples. Simultaneously, directing perturbations toward the neighboring clusters on the data manifold helps prevent deviations from the underlying data manifold, ensuring that adversarial examples remain realistic and meaningful.
> > > >
> > > > We acknowledge that these assumptions may not hold perfectly, as contrastive learning is performed without labeled data. However, this approach allows us to effectively generate adversarial examples using unlabeled data, supported by our experimental results.
> > > >
> > > > [1] T. Wang and P. Isola, ''Understanding Contrastive Representation Learning through Alignment and Uniformity on the Hypersphere,'' ICML'20

---

> > > > > ### Author Response · Authors · 2024-12-02
> > > > >
> > > > > > **Experiments on the prediction distribution for adversarial examples**
> > > > >
> > > > > The table below presents the experimental results comparing the prediction probabilities for adversarial examples generated using each method. Specifically, it shows the average prediction probabilities of the correct label (true label) and the incorrect label (misclassified label).
> > > > >
> > > > > The results demonstrate that in most cases, the adversarial examples generated by NCP-ACL are more effective, as evidenced by the higher prediction probabilities of the incorrect label compared to other methods. While these results are meaningful on their own, as explained in our first general response, the key contribution of NCP-ACL lies in its ability to generate adversarial examples that remain close to the data manifold. By utilizing these high-quality adversarial examples for adversarial training, NCP-ACL achieves superior robust accuracy, outperforming other SOTA methods. This advantage highlights the significance of NCP-ACL in advancing the field of adversarial contrastive learning.
> > > > >
> > > > > **Comparison of the prediction distribution for adversarial examples generated using NCP-ACL and SOTA methods.**
> > > > >
> > > > > This table compares the average prediction probabilities of correct labels (true labels) and incorrect labels (misclassified labels) for adversarial examples. All values are in %.
> > > > >
> > > > > |          | CIFAR-10               |                          | CIFAR-100               |                          | STL-10               |                          |
> > > > > |----------|-------------------------|--------------------------|--------------------------|--------------------------|-----------------------|--------------------------|
> > > > > |          | Correct Label   &nbsp;&nbsp;&nbsp;&nbsp;&nbsp;&nbsp;&nbsp;        | Incorrect Label &nbsp;&nbsp;&nbsp;&nbsp;&nbsp;&nbsp;&nbsp;           | Correct Label    &nbsp;&nbsp;&nbsp;&nbsp;&nbsp;&nbsp;&nbsp;        | Incorrect Label    &nbsp;&nbsp;&nbsp;&nbsp;&nbsp;&nbsp;&nbsp;       | Correct Label     &nbsp;&nbsp;&nbsp;&nbsp;&nbsp;&nbsp;&nbsp;     | Incorrect Label    &nbsp;&nbsp;&nbsp;&nbsp;&nbsp;&nbsp;&nbsp;       |
> > > > > | NCP-ACL  | **6.93**               | **61.10**               | **3.30**                | 33.84                   | 5.03                 | **74.35**               |
> > > > > | ACL      | 6.17                   | 61.05                   | 2.39                    | 34.85                   | **5.80**             | 62.50                   |
> > > > > | DeACL    | 5.37                   | 60.32                   | 1.56                    | **46.39**               | 4.60                 | 60.28                   |

---

> > > > > > ### Comment · Reviewer_E4Lt · 2024-12-03
> > > > > >
> > > > > > Thank you for your detailed explanations. Now, I can understand the empirical motivation of choosing K with the highest silhouette score, but still, I think there is no theoretical in-depth analysis to ensure your assumptions. Also, when I see the prediction distribution of the adversarial examples, it could be affected by the number of classes of tasks as shown in the results of CIFAR-100. To show your novelty and demonstrate that the proposed method supports your motivation, you need a more in-depth analysis and large-scale experiments with a larger number of classes. In response, I will maintain my score.

---

### Official Review · Reviewer_W6GT · 2024-11-04

**Soundness:** 2
**Presentation:** 3
**Contribution:** 2
**Rating:** 5
**Confidence:** 4

**Summary:**

The paper presents a novel method for adversarial contrastive learning that aims to improve the robustness of learned representations by leveraging the clustering structure inherent in the representation space formed during contrastive learning. NCP-ACL directs perturbations toward neighboring clusters (specifically the second nearest), hypothesizing that such perturbations are more likely to cross the decision boundary of a downstream classifier.

**Strengths:**

- The paper clearly explains the stages of NCP-ACL, from pre-training with SimCLR to the specific perturbation generation technique.
- The paper provides an extensive experimental analysis comparing NCP-ACL with existing state-of-the-art adversarial contrastive learning methods.
- The reported improvements in robust accuracy against strong adversarial attacks (e.g., PGD, Auto-Attack) are promising.
- The paper includes detailed ablation studies that examine the effects of different components, the number of clusters K, the balance parameter λ, and perturbation strategies.

**Weaknesses:**

- The relationship between the clustering in the representation space and the downstream classifier's decision boundary could be more clearly explained. The paper states that perturbations are likely to cross the boundary when directed toward neighboring clusters, but does this generalize across different downstream tasks? The assumption is not well motivated throughout the paper. Fig. 1 (b) for example shows that there exist clusters that are neighboring but not close to the decision boundary. The gray arrow case that is expected to "act as a positive sample in regular contrastive" needs further elaboration.
- The motivation for why targeting the second nearest cluster is particularly effective could be further elaborated. While the appendix empirically shows that perturbations directed toward this cluster lead to effective adversarial examples, the analysis is for the top-5 clusters and lacks a deeper theoretical or conceptual explanation that would strengthen the argument and provide more insight into the underlying mechanism.
- From a technical contribution, this work integrates clustering into the existing DeACL method. The novelty seems limited, and the method relies heavily on the quality of the clustering in the representation space. If the clusters are not well-formed, the perturbations generated may not be as effective in enhancing adversarial robustness.
- Decoupled adversarial contrastive learning for self-supervised adversarial robustness. In the related works section, the paper could better position itself within the adversarial CL literature.
- The experiments are conducted on CIFAR-10, CIFAR-100, and STL-10, which are relatively small-scale datasets. Including results from larger and more diverse datasets (e.g., ImageNet) would better showcase how the method performance scales.
- The paper discusses that the perturbed anchor sample $x + \delta$ may move closer to the perturbed negative set and further from the perturbed positive set when a sufficiently large number of negative samples is present. However, the statement does not seem to hold for recent works that adjust the direction of perturbations [1-3]. For example, targeted adversarial strategies, like TARO [1], demonstrate that simply perturbing samples without considering specific target directions or entropy measures can lead to suboptimal adversarial effects.
- Then lines 176-181 are also confusing since most of the literature deals with avoiding false negatives, i.e., examples that are considered negatives but have the same class as the anchor.
- There exists research that has explored neighbor-based perturbation strategies to enhance contrastive adversarial robustness, e.g., [4,5] and many works in graph learning. Lines 200-208 provide a proof-of-concept experiment that needs more depth to convincingly support the paper's claims, as it compares the effectiveness of the proposed perturbations solely with random noise. It would be best if this was an ablation study comparison with SoTA methods.
- To my understanding, SFL results reported in Table 3 are the same as the ones reported in Table 2 (CIFAR 10). The information could be condensed so that the paper presents both SFL and ALF results across all datasets.


[1] Kim, M., Ha, H., Son, S. and Hwang, S.J., 2024. Effective targeted attacks for adversarial self-supervised learning. Advances in Neural Information Processing Systems, 36.

[2] Wahed, M., Tabassum, A. and Lourentzou, I., 2022. Adversarial contrastive learning by permuting cluster assignments. arXiv preprint arXiv:2204.10314.

[3] Wang, X., Huang, Y., Zeng, D. and Qi, G.J., 2023. CaCo: Both positive and negative samples are directly learnable via cooperative-adversarial contrastive learning. IEEE Transactions on Pattern Analysis and Machine Intelligence, 45(9), pp.10718-10730.

[4] Jin, Y., Zhang, X., Lou, J., Ma, X., Wang, Z. and Chen, X., 2023. Explaining Adversarial Robustness of Neural Networks from Clustering Effect Perspective. In Proceedings of the IEEE/CVF International Conference on Computer Vision (pp. 4522-4531).

[5] Ko, C.Y., Mohapatra, J., Liu, S., Chen, P.Y., Daniel, L. and Weng, L., 2022, June. Revisiting contrastive learning through the lens of neighborhood component analysis: an integrated framework. In International Conference on Machine Learning (pp. 11387-11412). PMLR.

**Questions:**

- Does using silhouette scores to find the optimal number of clusters K involve manual interpretation?
- How does the method perform over existing clustering-based ACL methods?
- What does "act as a positive sample in regular contrastive" mean for grayed-out examples in Fig. 2?

---

> ### Author Response · Authors · 2024-11-25
>
> We sincerely appreciate the thoughtful feedback and valuable suggestions from Reviewer W6GT, which have greatly helped improve the quality of our work.
>
> >**Response to W1. Clarification on assumption**
>
> In our proposed method, we assume that the data manifold of the downstream task is sufficiently similar to that of the data used for contrastive learning. Without this assumption, performing contrastive learning on such data and using the pretrained encoder for downstream tasks would be inherently meaningless. We will clarify this point in future revisions of the manuscript.
>
> Based on this assumption, we demonstrated that generating perturbations that move away from the nearest cluster while simultaneously targeting a neighboring cluster in the representation space learned through contrastive learning is more likely to generate effective adversarial examples. Beyond this, we do not make any additional assumptions about the relationship between the clustering structure and the decision boundary of the downstream task classifier.
>
> Regarding generalization to different downstream tasks, the representation space learned through contrastive learning is task-agnostic, capturing the inherent structure of the data. Consequently, the clustering structure in the representation space reflects the underlying data distribution rather than being tied to any specific downstream task. This suggests that the effectiveness of perturbations is likely to generalize across downstream tasks, as long as the data manifold similarity assumption holds. Empirical evaluations across multiple datasets (e.g., CIFAR-10, CIFAR-100, and STL-10) support this claim, as the clustering-based perturbations consistently exhibit adversarial natures and lead to robust representations for downstream tasks.
>
> >**Response to W1&Q3. Clarification on grayed-out examples**
>
> Depending on the number of clusters and the clustering structure in the representation space, there exist neighboring clusters composed of data from the same class as the original sample and not located near the decision boundary. In such cases, the generated perturbations are directed toward regions where similar data points are located, resulting in samples (grayed-out examples) that share similar characteristics with the original samples.
>
> In NCP-ACL, the cosine similarity between these generated samples and the original samples is maximized. This process resembles the behavior of contrastive learning, where the cosine similarity between positive pairs is maximized. Therefore, we expect grayed-out examples to "act as a positive sample in regular contrastive learning," promoting alignment between similar data points in the representation space.
>
> >**Response to W2**
>
> We have addressed this topic in the second general response, where we provide a thorough explanation and supporting evidence for the rationale behind targeting the second nearest cluster. Please refer to the second general response.
>
> >**Response to W3**
>
> As highlighted in the first general response, our proposed method goes beyond simply integrating clustering into DeACL. It addresses a critical limitation of DeACL-the tendency to generate perturbations that deviate from the data manifold. By leveraging the well-clustered structure of the representation space learned through contrastive learning, NCP-ACL ensures that perturbations remain on the underlying data manifold, enhancing the adversarial robustness.
>
> Moreover, the clustering quality in the representation space is inherently guaranteed by the properties of contrastive learning. As demonstrated in [1], contrastive learning generally organizes similar data points positioned closely in the representation space, forming well-defined clusters. This clustering characteristic forms the foundation of our approach and minimizes the risk of poor clustering undermining the effectiveness of our method. Far from being a limitation, the reliance on clustering quality is a key strength of NCP-ACL, as it utilizes the intrinsic structure of representations of contrastive learning for effective adversarial training.
>
> [1] T. Wang and P. Isola, ''Understanding Contrastive Representation Learning through Alignment and Uniformity on the Hypersphere,'' ICML'20
>
> >**Response to W4**
>
> We acknowledge the importance of explicitly discussing DeACL in the related works section to better emphasize the progression and unique aspects of our approach. In future revisions, we will enhance this section with a more comprehensive discussion of DeACL.

---

> > ### Author Response · Authors · 2024-11-25
> >
> > > **Response to W5**
> >
> > We agree that incorporating results from larger and more diverse datasets like ImageNet would offer valuable insights into the scalability and generalizability of our method. However, a significant challenge in adversarial contrastive learning research is the high computational cost, which has led to a focus on relatively small-scale datasets such as CIFAR-10, CIFAR-100, and STL-10. For instance, ADVCL, one of the most computationally intensive adversarial contrastive learning methods, requires over 70 hours to train on STL-10 using a single NVIDIA 3090 RTX GPU. As a result, most existing studies in adversarial contrastive learning also limit their evaluations to these datasets. In future work, we plan to evaluate NCP-ACL on larger datasets like ImageNet to further demonstrate its scalability and robustness in diverse settings.
> >
> > > **Response to W6**
> >
> > [1] aims to improve adversarial robustness for positive-only self-supervised methods but performs significantly worse than standard adversarial contrastive learning approaches that incorporate negative pairs. Specifically, it falls behind the SOTA methods by 5\% on the PGD-20 attack and over 10\% on AutoAttack, limiting its effectiveness.
> >
> > [2] includes a selection process for negative pairs but still relies on maximizing contrastive loss with these negative pairs, resulting in similar issues. In addition, it underperforms SOTA methods by more than 10\% on the PGD-20 attack.
> >
> > Finally, [3] is not an adversarial contrastive learning method but rather a strategy to improve standard contrastive learning. Its goals and scope differ from our work and do not address generating adversarial examples for robust training.
> >
> > > **Response to W7**
> >
> > As the reviewer pointed out, there has been considerable research in contrastive learning aimed at avoiding false negatives, as demonstrated in works like [1] and [2]. We will add these references to clarify the context in lines 176–181.
> >
> > The main point we intended to convey in lines 176-181 is that existing adversarial contrastive learning methods lack explicit strategies to address the false negative issue. As a result, this limitation leads to the generation of ineffective adversarial examples, which ultimately compromises the robustness of the learned representations.
> >
> > [1] J. Robinson, C.-Y. Chuang, S. Sra, and S. Jegelka, ''Contrastive learning with hard negative samples,'' ICLR 2021
> >
> > [2] Y. Kalantidis, M. B. Sariyildiz, N. Pion, P. Weinzaepfel, and D. Larlus, ''Hard negative mixing for contrastive learning,'' NeurIPS 2020
> >
> > > **Response to W8-1. Research on neighbor-based perturbation generation strategies**
> >
> > Thank you for pointing out the research on neighbor-based perturbation strategies in graph learning. We will look into these studies as valuable references for future work. However, [4] is a supervised adversarial training method that utilizes labeled data to enhance robustness against adversarial attacks. Since our work focuses on learning robust models using unlabeled data, [4] falls outside the scope of this study.
> >
> > Regarding IntNaCl [5], it generates perturbations by maximizing the contrastive loss, similar to other adversarial contrastive learning methods,  and does not employ neighbor-based perturbation strategies. We will add IntNaCl as a reference in the related works section to provide a more comprehensive context. Thank you for bringing these interesting studies to our attention.
> >
> > > **Response to W8-2. Comparison of the effectiveness of perturbations generated by NCP-ACL and the SOTA methods**
> >
> > The proof-of-concept experiment comparing random noise with perturbations directed toward the second-nearest cluster was intended to validate whether perturbations targeting the second-nearest cluster have an adversarial impact. Random noise served merely as a baseline to confirm that the perturbations we generated exhibit adversarial natures. It was not designed to demonstrate the superiority of our method. In the first general response, we present the comparative experiments evaluating the effectiveness of perturbations crafted by NCP-ACL and the SOTA methods. Please refer to it.

---

> ### Author Response · Authors · 2024-11-25
>
> > **Response to W9**
>
> We consolidated the results of both standard linear finetuning (SLF) and adversarial linear finetuning (ALF) into a single table across all datasets. This table provides a comprehensive comparison of the effects of these finetuning strategies on encoders trained using adversarial contrastive learning methods.
>
> The table demonstrates that NCP-ACL achieves the best overall performance in terms of RA and AA when considering both SLF and ALF altogether, highlighting the inherent robustness of the representations learned by NCP-ACL.
>
> **Performance comparison of adversarial contrastive learning methods under different fine-tuning strategies across multiple datasets.**
> (The best and the second-best results highlighted in **bold** and *italic*, respectively)
>
> | Method            | Finetuning Method                   | CIFAR-10          |                  |                  | CIFAR-100        |                  |                  | STL-10           |                  |                  |
> |--------------------|-------------------------------------|-------------------|------------------|------------------|------------------|------------------|------------------|------------------|------------------|------------------|
> |                    |                                     | SA (%)    &nbsp;&nbsp;&nbsp;&nbsp;&nbsp;&nbsp;&nbsp;&nbsp;&nbsp;       | RA (%)    &nbsp;&nbsp;&nbsp;        | AA (%)     &nbsp;&nbsp;&nbsp;      | SA (%)      &nbsp;&nbsp;&nbsp;&nbsp;&nbsp;&nbsp;&nbsp;&nbsp;&nbsp; &nbsp;&nbsp;&nbsp;&nbsp;     | RA (%)    &nbsp;&nbsp;&nbsp;       | AA (%)    &nbsp; &nbsp; &nbsp;        | SA (%)     &nbsp; &nbsp; &nbsp; &nbsp; &nbsp; &nbsp; &nbsp;       | RA (%)      &nbsp; &nbsp; &nbsp;      | AA (%)    &nbsp; &nbsp; &nbsp;        |
> | **ADVCL**          | Standard Linear Finetuning (SLF)   | 80.85             | 50.45            | 42.57            | 48.34            | 27.67            | 19.78            | n/a              | n/a              | n/a              |
> | **DeACL**          |                                     | 80.17             | 53.95            | 45.31            | 52.79            | *30.74*          | 20.34            | 80.87            | 65.86            | 58.50            |
> | **DYNACL**         |                                     | 75.38             | 48.48            | 45.56            | 45.85            | 22.76            | 19.43            | 69.61            | 48.76            | 47.17            |
> | **DYNACL++**       |                                     | 79.79             | 49.37            | 47.05            | **53.93**        | 21.97            | 19.71            | 70.97            | 49.91            | 47.80            |
> | **DYNACL-AIR++**   |                                     | *81.86*           | 49.01            | 47.12            | *53.90*          | 23.20            | 20.69            | 71.36            | 50.23            | 48.42            |
> | **NCP-ACL (ours)** |                                     | **82.02**         | *54.06*          | *49.22*          | 51.00            | **32.01**        | **23.32**        | **84.06**        | *68.11*          | *63.55*          |
> | **ADVCL**          | Adversarial Linear Finetuning (ALF)| 80.00             | 51.93            | 44.65            | 47.08            | 27.99            | 20.19            | n/a              | n/a              | n/a              |
> | **DeACL**          |                                     | 78.81             | **54.61**        | 46.09            | 42.74            | 28.87            | 20.77            | 80.65            | 66.92            | 57.87            |
> | **DYNACL**         |                                     | 72.89             | 49.66            | 46.01            | 43.63            | 25.57            | 20.79            | 67.83            | 50.85            | 48.43            |
> | **DYNACL++**       |                                     | 78.82             | 51.18            | 48.46            | 51.35            | 26.88            | 22.74            | 69.68            | 51.35            | 48.50            |
> | **DYNACL-AIR++**   |                                     | 79.59             | 51.77            | 48.66            | 51.91            | 27.10            | *23.09*          | 70.55            | 52.08            | 49.84            |
> | **NCP-ACL (ours)** |                                     | 80.40             | **54.61**        | **49.23**        | 44.98            | 30.54            | 22.66            | *83.88*          | **70.29**        | **64.81**        |

---

> > ### Author Response · Authors · 2024-11-25
> >
> > > **Response to Q1**
> >
> > No, using silhouette scores to determine the optimal number of clusters $K$ does not involve manual interpretation in our method. The score is computed automatically for various values of $K$, and the value that maximizes the silhouette score is selected as the optimal number of clusters. This process is fully automated and does not require manual intervention.
> >
> > > **Response to Q2**
> >
> > We assumed that the reviewer was referring to [4] and [5] as the clustering-based ACL since there were no previously presented clustering-based ACL methods in the context of our work. If the reviewer intended to reference other methods, we would appreciate it if they could provide further clarification.
> >
> > The reviewer-provided [4] is excluded from comparison as it focuses on supervised adversarial training, which falls outside the scope of our study. Regarding [5], we could not perform direct experiments due to the unavailability of pretrained model weights or implementation code. However, the reported results in [5] provide robust accuracy under the PGD attack with
> > $\epsilon=0.002$. Comparing this to NCP-ACL, which was evaluated under a much stronger $\epsilon=0.03$, we observed that NCP-ACL achieved over 5\% higher robust accuracy on CIFAR-10, highlighting its superior performance despite the more challenging attack settings.

---

> > > ### Comment · Reviewer_W6GT · 2024-11-29
> > > **Thank you for the thorough rebuttal**
> > >
> > > Thank you for the rebuttal. I have raised my score due to the comprehensive rebuttal, however, it is worth noting that several of my primary concerns remain after reading all reviews and author responses. I agree with other reviewers that the lack of theoretical depth about why the second-nearest cluster is targeted remains a key limiting factor for the soundness of the approach. The novelty of leveraging clustering still feels incremental compared to DeACL and other related works mentioned by reviewers, and larger-scale experiments are needed to solidify the paper's claims.

---

### Comment · Area_Chair_ThiD · 2024-11-22

Dear Authors and Reviewers,

The discussion phase has passed 10 days. If you want to discuss this with each other, please post your thoughts by adding official comments.

Thanks for your efforts and contributions to ICLR 2025.

Best regards,

Your Area Chair

---

### Author Response · Authors · 2024-11-25
**General Response 1. Comparison of Perturbation Effectiveness: NCP-ACL and SOTA Methods**

We thank the reviewers for their positive feedback and constructive comments. This response focuses on addressing the primary concern: a lack of comparative experiments evaluating the effectiveness of perturbations generated by our proposed NCP-ACL and SOTA methods in adversarial training. We provide an analysis comparing these perturbations in terms of their adversarial impact, deviation from the data manifold, and their overall effect on standard and robust accuracy.

It is important to note that strong adversarial perturbations are not always beneficial for adversarial training. While such perturbations may exhibit high adversarial impact, they can significantly deviate from the underlying data manifold. This deviation can bring severe distortions in the representation space learned during contrastive learning, ultimately degrading the standard and robust accuracy of models used in downstream tasks. Therefore, the effectiveness of perturbations should be evaluated not only in terms of their adversarial impact but also by considering their deviation from the data manifold.

The Table 1 demonstrates that DeACL generates perturbations with the highest adversarial impact, achieving the largest Prob. Dec. and Attk. Succ. values. However, as shown in the Table 2, DeACL's adversarial examples are farthest from the training data in the representation space. In other words, adversarial examples of DeACL are more likely to deviate significantly from the underlying data manifold. These findings are consistent with experimental results presented in our paper (lines 341-349). Our NCP-ACL generates adversarial examples closest to the data manifold, increasing the likelihood of producing on-manifold adversarial examples. This results in the NCP-ACL's superior performance compared to DeACL and other ACL-based methods.
The proposed NCP-ACL tries to generate perturbations far away from the nearest cluster (to be adversarial) and toward the second nearest cluster (to be close to the training data).

We will add these remarks with further clarifications in the final version of the manuscript.

**Table 1. Comparison of the adversarial impact of perturbations generated using NCP-ACL and SOTA methods.**

|               | CIFAR-10          |                | CIFAR-100        |                | STL-10           |                |
|---------------|-------------------|----------------|------------------|----------------|------------------|----------------|
|               | Prob. Dec.        | Attk. Succ.    | Prob. Dec.       | Attk. Succ.    | Prob. Dec.       | Attk. Succ.    |
| **NCP-ACL**   | 45.88             | 62.37          | 46.68            | 84.67          | 52.20            | 58.91          |
| **ACL**       | 49.99             | 66.27          | 40.94            | 77.99          | 54.83            | 61.12          |
| **DeACL**     | **69.90**         | **89.06**      | **54.31**        | **94.95**      | **80.44**        | **89.12**      |


**Details on Table 1.** In this table, ACL represents perturbations generated by maximizing the contrastive loss, the strategy used by SOTA methods such as AdvCL, DynACL, DynACL++, DynACL-AIR++, and IntNaCl.

**Table 2. Comparison of Euclidean distances between adversarial examples generated using adversarial contrastive learning methods and on-manifold data in representation space.**

| Method   | CIFAR-10 | CIFAR-100 | STL-10 |
|----------|----------|-----------|--------|
| **NCP-ACL** | **2.90**  | **3.03**   | **5.88** |
| **ACL**      | 3.16     | 3.58      | 6.03   |
| **DeACL**    | 4.12     | 4.21      | 6.48   |

**Details on Table 2.** To measure how far the generated adversarial examples deviate from the underlying manifold, we calculated the minimum Euclidean distances between the adversarial examples and 1,000 randomly sampled data points from the downstream data distribution in the representation space.

---

> ### Author Response · Authors · 2024-11-25
> **General Response 2. Rationale for Targeting the Second-Nearest Cluster in NCP-ACL**
>
> We recognize the need to clarify why perturbations are specifically directed toward the second-nearest cluster, and this response aims to address that question.
>
> The NCP-ACL method is designed to generate perturbations that exhibit adversarial characteristics while remaining aligned with the data manifold. By directing perturbations away from the nearest cluster, which consists of data similar to the original samples, the method achieves an adversarial effect. Simultaneously, it guides the perturbations toward a neighboring cluster on the data manifold, ensuring that the generated adversarial examples maintain their alignment with the underlying data manifold.
>
> As a result, the specific choice of which neighboring cluster to target (e.g., the second-nearest or third-nearest) is not a critical factor, as long as the perturbation remains on the data manifold and retains an adversarial nature. This is supported by our experimental results, which show that increasing the value of $n$ (i.e., targeting clusters farther away) does not lead to significant changes in performance. These findings suggest that the effectiveness of the method remains consistent despite variations in the choice of the targeted cluster, as long as the core concept of adversarial perturbation and on-manifold alignment are preserved. This finding led us to adopt a simpler approach: generating perturbations toward the second-nearest cluster.
>
> **The impact of targeting different neighboring clusters for adversarial contrastive learning.**
>
> |   | **CIFAR-10**     |                  |                  | **CIFAR-100**    |                  |                  | **STL-10**      |                  |                  |
> |--------------------------|------------------|------------------|------------------|------------------|------------------|------------------|------------------|------------------|------------------|
> |                          | SA (%)          | RA (%)   &nbsp;&nbsp;&nbsp;       | AA (%)     &nbsp;&nbsp;&nbsp;&nbsp;      | SA (%)           | RA (%)    &nbsp;&nbsp;&nbsp;        | AA (%)  &nbsp;&nbsp;&nbsp;          | SA (%)           | RA (%)   &nbsp;&nbsp;&nbsp;        | AA (%)     &nbsp;&nbsp;&nbsp;      |
> | n=2 (NCP-ACL)           | 82.02           | 54.06           | 49.22           | 51.00            | 32.01            | 23.32            | 84.06            | 68.11            | 63.55            |
> | n=3                     | 81.73           | 54.41           | 49.93           | 51.06            | 32.32            | 23.64            | 83.50            | 68.38            | 64.25            |
> | n=4                     | 81.87           | 53.91           | 49.85           | 51.16            | 31.91            | 23.63            | 83.47            | 68.01            | 63.67            |
> | n=5                     | 82.09           | 54.11           | 49.72           | 51.14            | 31.61            | 23.41            | 83.17            | 67.79            | 63.39            |
> | n=6                     | 82.21           | 53.92           | 49.86           | 50.97            | 31.59            | 23.42            | 83.07            | 67.51            | 63.51            |
> | n=7                     | 82.13           | 54.13           | 49.85           | 51.28            | 31.81            | 23.51            | 83.24            | 67.40            | 63.15            |
> | n=8                     | 82.34           | 54.20           | 49.84           | 51.25            | 31.68            | 23.13            | 83.31            | 67.79            | 63.80            |
> | n=9                     | 82.18           | 54.14           | 49.73           | 51.17            | 31.32            | 23.07            | 83.09            | 67.94            | 63.56            |
> | n=10                    | 82.21           | 53.96           | 49.80           | 51.43            | 31.75            | 23.35            | 83.16            | 67.12            | 63.09            |

---

### Comment · Area_Chair_ThiD · 2024-11-30
**Please double check the authors' responses and revision and update the rating if suitable**

Dear Reviewers,

Thanks for the engagement during the discussion. Because the final discussion deadline is fast approaching, please double-check the authors' responses and revisions and update the rating if you feel possible.

Best regards,

Area Chair

---

### Meta-Review · Area_Chair_ThiD · 2024-12-20

**Metareview:**

The key idea of this paper is quite interesting: first finding that the effectiveness of adversarial contrastive learning is influenced by the composition of positive and negative examples in a minibatch, then proposing a novel approach to adversarial contrastive learning, where adversarial perturbations are generated based on the clustering structure of the representation space learned through contrastive learning. Reviewers also like this idea and agree on its novelty. However, the current evaluation (empirical or theoretical) is still weak and cannot fully support the claim made in the current version. Another round would be beneficial for further improving this paper's quality.

**Additional Comments On Reviewer Discussion:**

More solid support for the claims is also mentioned in reviews and rebuttals.

---

### Decision · Program_Chairs · 2025-01-22

Reject